# Genome skimming and microsatellite analysis reveal contrasting patterns of genetic diversity in a rare sandhill endemic (*Erysimum teretifolium*, Brassicaceae)

José Carlos del Valle[1☯], Julie A. Herman[2☯], Justen B. Whittall[2]*

1 Department of Molecular Biology and Biochemical Engineering, Pablo de Olavide University, Seville, Spain,
2 Department of Biology, Santa Clara University, Santa Clara, CA, United States of America

☯ These authors contributed equally to this work.
* jwhittall@scu.edu

## Abstract

Barriers between islands often inhibit gene flow creating patterns of isolation by distance. In island species, the majority of genetic diversity should be distributed among isolated populations. However, a self-incompatible mating system leads to higher genetic variation within populations and very little between-population subdivision. We examine these two contrasting predictions in *Erysimum teretifolium*, a rare self-incompatible plant endemic to island-like sandhill habitats in Santa Cruz County, California. We used genome skimming and nuclear microsatellites to assess the distribution of genetic diversity within and among eight of the 13 remaining populations. Phylogenetic analyses of the chloroplast genomes revealed a deep separation of three of the eight populations. The nuclear ribosomal DNA cistron showed no genetic subdivision. Nuclear microsatellites suggest 83% of genetic variation resides within populations. Despite this, 18 of 28 between-population comparisons exhibited significant population structure (mean $F_{ST} = 0.153$). No isolation by distance existed among all populations, however when one outlier population was removed from the analysis due to uncertain provenance, significant isolation by distance emerged ($r^2 = 0.5611$, $p = 0.005$). Population census size did not correlate with allelic richness as predicted on islands. Bayesian population assignment detected six genetic groupings with substantial admixture. Unique genetic clusters were concentrated at the periphery of the species' range. Since the overall distribution of nuclear genetic diversity reflects *E. tereifolium*'s self-incompatible mating system, the vast majority of genetic variation could be sampled within any individual population. Yet, the chloroplast genome results suggest a deep split and some of the nuclear microsatellite analyses indicate some island-like patterns of genetic diversity. Restoration efforts intending to maximize genetic variation should include representatives from both lineages of the chloroplast genome and, for maximum nuclear genetic diversity, should include representatives of the smaller, peripheral populations.

**Data Availability Statement:** All relevant data are within the manuscript and its Supporting Information files.

**Funding:** This research was supported by a Section VI grant from the California Department of Fish and Wildlife (https://www.wildlife.ca.gov/) to JBW (Grant #P1182012). JAH was supported by an ALZA Corporation Scholarship. The funders had no role in study design, data collection and analysis, decision to publish, or preparation of the manuscript.

**Competing interests:** The authors have declared that no competing interests exist.

## Introduction

The isolating nature of islands offers a unique window into evolution. Islands are often separated by barriers to migration which are predicted to produce unique genetic footprints [1–4]. Island species are characterized by having restricted gene flow among islands with a gradient of decreasing gene flow as islands become more distantly separated. This creates genetic structure among islands where most of the genetic variation resides among islands rather than within islands (Hypothesis #3 in [4]). This prediction is supported by several empirical studies [5,6], with noteworthy exceptions [7,8]. In addition to geographical isolation, life-history characters, such as fruit type, have been identified as important factors in determining the distribution of genetic variation within and among islands. For example, species with fleshy fruits are suited for frequent dispersal by animals, favoring genetic cohesion in otherwise fragmented landscapes like islands [9].

Comparable predictions regarding the distribution of genetic diversity within and among islands can be found in Wright's island model of population genetics (e.g., isolation by distance) [10]. This model is based on geographically distinct populations separated by barriers to gene flow but can also be applied to continental habitats that are island-like. However, investigations into the distribution of genetic variation in island-like habitats are relatively rare given the diversity of naturally patchy, edaphically unique habitats (reviewed in [11]). Complementary to this line of inquiry are a wealth of population genetic studies of species where distributions are characterized by a central, large, relatively contiguous meta-population surrounded by several peripheral, smaller, isolated populations [12]. For example, in a meta-analysis of 134 population genetic studies of species with a clear center-margin population distinction, Eckert et al. found lower genetic variation in marginal (peripheral) populations compared to central populations (64.2% of studies) and higher genetic subdivision in comparisons between central and peripheral populations (70.2% of studies), reflecting the island-like nature of most marginal populations [12].

The genetic predictions for species on islands and in island-like habitats must also account for life-history traits that can directly affect gene flow, such as mating system [9,13] and seed dispersal [14]. In terms of mating system, many island species are self-compatible [15], which greatly affect the distribution of genetic diversity [16]. Outside of islands, mating system, specifically outcrossing rate, is repeatedly the single best predictor of the distribution of genetic diversity in several plant population genetic meta-analyses [17–19]. Outcrossing homogenizes the genetic differences among populations and maintains this variation within populations over long periods. Thus, most genetic variation is harbored within populations for outcrossing lineages [20,21]. Meanwhile, selfing reduces the effective population size and increases the impact of drift, which collectively leads to the accumulation of genetic differences among populations and increased homogeneity within them. A self-incompatible species restricted to island-like habitats would provide a rare opportunity to test the contrasting predictions posing mating systems against island-like habitats as the major factor determining the distribution of genetic diversity.

*Erysimum teretifolium* (Brassicaceae) is a self-incompatible plant endemic to the Zayante sandhills of Santa Cruz County, California (Fig 1A). The Zayante sandhills are island-like, xeric habitats separated by relatively mesic redwood (*Sequoia sempervirens*) and mixed evergreen forests. These Miocene-era, uplifted sea floors are characterized by depauperate levels of macronutrients and extremely low water holding capacity (J. McGraw, personal communication). Several animals and plants have adapted to these unique island-like habitats including the Mt. Hermon June Beetle (*Polyphylla barbata*), Zayante band-winged grasshopper (*Trimerotropis infantilis*), Santa Cruz kangaroo rat (*Dipodomys venustus venustus*), and the subject of

A.

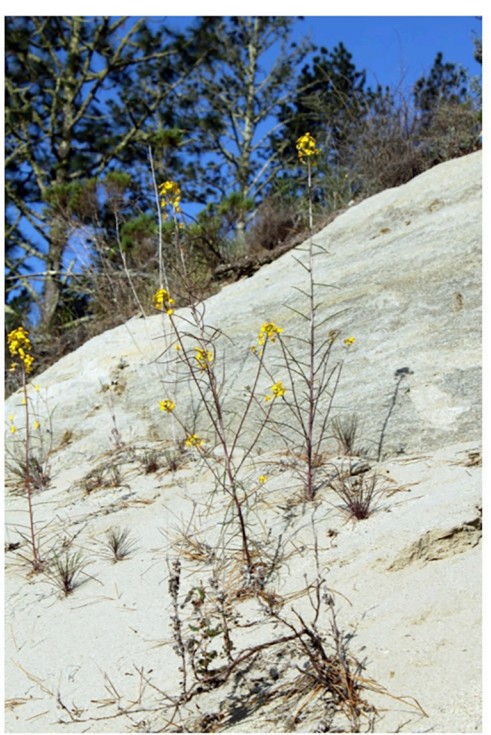

B.

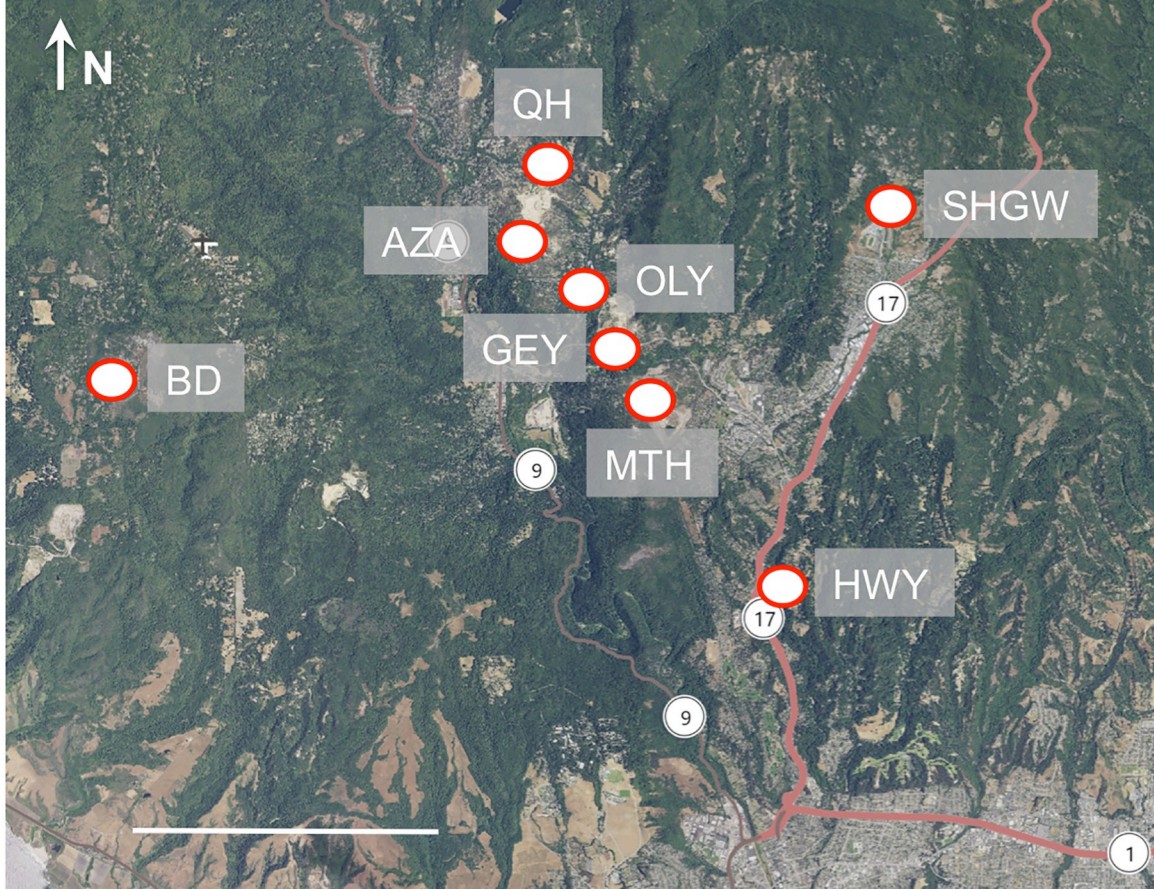

**Fig 1. The Ben Lomond wallflower and location of the sampled populations.** *Erysimum teretifolium* in sandhill parkland habitat at Mount Hermon (A) and the geographic location and abbreviations for the eight studied populations in the Zayante sandhills in Santa Cruz County, California, USA (B). Base map provided by the USGS National Map Viewer (http://viewer.nationalmap.gov/viewer/). Scale bar, 5km.

this study, the Ben Lomond Wallflower (*Erysimum teretifolium*). In reference to the high concentration of endemism and the island-like nature of the Zayante sandhills, Peter Raven once referred to this region as "the Galápagos Islands of Santa Cruz County," (P. Raven, Missouri Botanical Garden, http://www.landtrustsantacruz.org/sandhills/dr_raven.htm). Originally consisting of 2,535 hectares [22–25], this naturally patchy habitat is threatened by the sand quarrying industry and residential development [26]. Now, it is estimated that only ~1,600 hectares remain [24]. Although the concentration of biodiversity and habitat distribution in the Zayante sandhills is remarkably island-like, it remains unclear whether the distribution of genetic diversity among its inhabitants match that predicted for oceanic island endemics or is more consistent with the obligate outcrossing mating system of many of the inhabitants including *E. teretifolium* [27].

Erysimum teretifolium occupies an open subset of the Zayante sandhills dotted with Ponderosa pines (*Pinus ponderosa*) known as sand parkland [24]. As a unique subtype of sandhill habitat, sandhill parkland of the Zayante sandhills was originally estimated to occupy approximately 240 hectares, but is currently known from less than 80 hectares [24,28]. Paradoxical for an island-like endemic plant, this perennial species is self-incompatible and attracts a diversity of pollinators from the orders Coleoptera, Diptera, Hymenoptera, and Lepidoptera [27]. In experimental crosses, outcross pollinations produce approximately $6.5 \times$ more seeds than self pollinations, and most self pollinations produced no seeds consistent with a self-incompatible mating system [27]. Neither the fruits (7–11 cm long siliques), nor the seeds (< 0.5 mg each) exhibit obvious long-distance dispersal mechanisms suited to zoochory or anemochory, although some seeds have a very small (< 1 mm) wing on the distal margin [29]. As a member of the *E. capitatum* alliance [29], it is a putative hexaploid with unknown parentage (2n = 36; [29]).

In 1998, 16 remaining populations of *E. teretifolium* were identified [28]. Half of these reported populations were on private land and thus, were inaccessible for this study. Four to six populations form a central cluster and are separated by as little as 1 km, while other outlying populations are separated by as many as 12 km (Table 1). These populations range in size from 35 individuals to over 1,000 individuals [30]. As a result of habitat destruction and small population census sizes, the species was listed as federally endangered in 1994 [31]. A historically large population (~ 2,000 individuals) at the Bonny Doon Ecological Reserve has recently spiraled into near extirpation with less than 10 reproductive adults between 2013–2019 due to competition from encroaching sandhill chaparral and reproductive failure [30]. In hopes of rescuing this population, locals may have supplemented it with seeds from a nearby, more accessible population at Quail Hollow County Park (J. McGraw & V. Haley, personal communication). The Bonny Doon Ecological Reserve population is now the focus of a major reintroduction effort (T. Kasteen, CA Fish and Wildlife Service, personal communication). Because of the concentration of rare endemics in the Zayante sandhills and unique edaphic regions in general, *E. teretifolium* could be a model for the distribution of genetic diversity within and among populations in continental island-like habitats, which could help guide future conservation efforts including where to source material for restoration efforts in the Zayante sandhills and beyond.

In this study, we examine the distribution of genetic diversity within and among eight island-like populations of *E. teretifolium* (Fig 1B) using a genome skimming approach

**Table 1. Population location, census size, sampling, and microsatellite genetic diversity.**

| Population | Abbrev. | Geographic location | Census size | Total no. samples | No. samples from captive breeding population | Total no. microsatellite fragments scored | No. fixed microsatellite fragments within population |
|---|---|---|---|---|---|---|---|
| Azalea Rd. | AZA | 37˚04'21.97"N 122˚03'42.39"W | 450 | 22 | n/a | 15 | 6 |
| Bonny Doon Ecological Reserve | BD | 37˚02'24.88"N 122˚05'20.75"W | 1000* | 25 | 25 | 12 | 6 |
| Geyer Quarry/ Randall Morgan Preserve | GEY | 37˚04'51.27"N 122˚00'18.64"W | 500 | 42 | 24 | 15 | 4 |
| Highway 17 | HWY | 37˚01'29.90"N 122˚01'35.30"W | 35 | 11 | n/a | 12 | 7 |
| Mount Hermon/ Hansen Quarry | MTH | 37˚02'55.52"N 122˚02'57.10"W | 150 | 24 | n/a | 15 | 6 |
| Olympia Quarry/ San Lorenzo Valley Water District | OLY | 37˚04'51.27"N 122˚00'18.64"W | 650 | 43 | 18 | 15 | 4 |
| Quail Hollow County Park | QH | 37˚04'51.27"N 122˚00'18.64"W | 750 | 42 | 18 | 15 | 6 |
| Sandhill Rd. and Glenwood Dr. | SHGW | 37˚04'51.92"N 122˚00'18.41W | 50 | 24 | n/a | 12 | 5 |

* Bonny Doon census size estimated from the average before the rapid population decline in approximately 2013.

complemented with nuclear microsatellites to compare the contrasting effects of geography and mating system on the distribution of genetic diversity. Genome skimming relies on next-generation sequencing technology to efficiently capture the majority of the chloroplast genome, the nuclear ribosomal cistron (including the hypervariable internal transcribed spacer and external transcribed spacer regions; hereafter referred to as nrDNA), and sometimes even portions of the mitochondrial genome [32]. We apply this data in combination with four nuclear microsatellite markers to test the contrasting predictions from its island-like habitat versus the self-incompatible mating system on the distribution of genetic diversity. The island model predicts that most genetic variation will be found *among* populations and strong genetic structure will exist among isolated populations because of barriers to gene flow. Alternatively, the self-incompatible mating system should maintain high levels of genetic variation *within* populations with very little variation attributed to among-population comparisons. The conservation implications of these results will guide how to sample source material for future reintroduction and restoration efforts.

## Materials and methods

### Genome sizing

In order to confirm the hexaploid nature of *E. teretifolium* (hereafter, ERTE), one individual from four geographically distinct populations (Bonny Doon, Geyer, Olympia, and Quail Hollow) was grown in a greenhouse for genome size estimation using flow cytometry (Benaroya Research Institute at Virginia Mason, Seattle, WA). One hundred mg of fresh leaf tissue was stored on ice for approximately 24 hours until it could be homogenized. Cells were initially lysed in Galbraith's buffer [33] and nuclei were stained with propidium iodide and treated with RNase. Measurements were performed on a Becton Dickinson FACS flow cytometer. Genome size was estimated in comparison to chicken red blood cells (2C = 2.33 pg) since plant standards were not readily available. Four measurements were taken per sample. Dolezel

et al.'s method [34] was used to convert nuclear mass into base pairs (978 million base pairs per pg of diploid nuclear DNA).

## Sampling

Leaf samples were collected from eight ERTE populations (11–43 individuals per population; Fig 1B, Table 1). This work was permitted under Agreement #P1182012 00 with the California Department of Fish and Wildlife. All field sites were accessed with permission from the appropriate authorities (Lee Summers at Quail Hollow County Park, Betsy Herbert at San Lorenzo Valley Water District, Valerie Haley at Mount Hermon, Dr. Jodi McGraw at Azalea Rd. and Sandhill Roadd/Glenwood Drive, Lynn Overtree at the Randall Morgan Preserve and Terris Kasteen at the Bonny Doon Ecological Reserve). An additional 27 samples were collected from both subpopulations of the single known location of *E. capitatum* ssp. *angustatum* (hereafter, ERCAAN) at the Antioch Dunes National Wildlife Reserve, California. This species, as with the all the taxa in the *E. capitatum* alliance, is also a hexaploid (2n = 36; [29]).

This work was conducted under the same permit number and field sites were accessed with permission from Susan Euing at the U.S. Fish and Wildlife Service, Antioch Dunes National Wildlife Reserve. This species was included to compare the genetic variation within and among species and provided a reference for how much variability to expect between closely related species. ERCAAN and ERTE both belong to the recent radiation of the *E. capitatum* alliance in western North America, yet the two are geographically and morphologically distinct [29].

For both species, individuals separated by a minimum of two meters were sampled broadly throughout each population. Because of conservation concerns at four ERTE populations (Bonny Doon, Geyer, Olympia, and Quail Hollow), some or all of the samples were derived from a captive breeding population established by Dr. Ingrid Parker at University of California, Santa Cruz (Table 1). For the captive breeding populations, we sampled from 9–13 maternal families originating from two to three geographically distinct patches separated by 10–62 meters. Leaf samples were stored at -20C until DNA could be extracted using the NucleoSpin Plant II kit with lysis buffer PL1 (Macherey-Nagel, Düren, Germany).

## Genome skimming methodology

One μg of DNA from each of 12–25 individuals per population (a subset of all of the samples) were pooled after quantification by Qubit (Thermo-Fisher, Waltham, MA). Genomic DNA library preparation was completed according to the manufacturer's protocol by Novogene (Beijing, China). These eight samples (populations) plus two outgroup samples (ERCAAN) were indexed and then sequenced on a single lane of Illumina HiSeq2000 (San Diego, USA) producing 250bp paired-end reads.

Data was processed, assembled, and analyzed primarily in Geneious v.8.1.6 (Biomatters Ltd., Auckland, New Zealand). Initially, *fastq* files were trimmed using the default settings in Geneious (i.e. 3' and 5' ends of sequences with more than a 5% chance of an error per base were removed). Next, we assembled the sequences using *Arabidopsis thaliana* genomes as the references (GenBank accession numbers NC_000932 and NC_001284 for chloroplast and mitochondrial genomes, respectively). For the nrDNA cistron, we created a mosaic reference substituting *Erysimum capitatum* sequences for the hypervariable internal transcribed spacer region (GenBank accession numbers AY254534, DQ357540, KJ417987 and KJ417988), but relying on the *Arabidopsis* nrDNA (GenBank accession numbers X52320 and X16077) for the remainder since no complete sequences of the large and small subunits were available for *Erysimum*. We conducted assemblies of the chloroplast genome, mitochondrial genome and

nrDNA cistron separately using the Geneious assembler under default settings with medium-low sensitivity and 10 iterations [35]. A consensus sequences for each sample was extracted requiring a 75% match to the reference and 5x minimum coverage. Annotations were transferred using a 75% similarity cutoff to the reference genome. DNA sequence alignments were created using the MAFFT plugin [36] with default settings, followed by visual inspection and manual adjustments when necessary. Phylogenetic analyses were conducted using RAxML with the GTR+CAT approximation of the GTR+G model of nucleotide evolution with estimate of proportion of invariable sites and 1,000 bootstrap replicates [37]. The resulting trees were visualized using FigTree v.1.4.2 [38].

Genetic diversity in the chloroplast genome was examined using the Discriminant Analysis of Principal Components (DAPC) [39,40]. Ambiguities were not included in the analysis since DAPC treats them as additional characters, not as the ambiguities that they are (following the command "SNP<-DNAbin2genind"). We could not use DAPC for the nrDNA since 58 of 59 SNPs contained ambiguities for at least one of the samples. We compared DAPC analyses using *a priori* defined groups. First, we used two partitions (ERCAAN vs. ERTE) and then we expanded to three partitions based on the two distinct ERTE lineages (ERCAAN vs. GEY, OLY, QH, SHGW and MTH vs. AZA, BD, and HWY) recovered in the phylogenetic analysis of the chloroplast genome. The number of principal components was set according to alpha-score optimization (i.e., trade-off between power of discrimination and overfitting) [41]. DAPC analysis was implemented in R v.3.2.3 [42] using the package 'adegenet' v.2.0.0 [39].

## Microsatellite methodology

After an initial survey of the 10 nuclear microsatellite loci developed from the European *E. mediohispanicum* [43], we selected four of the most promising loci for further analysis. PCR was performed in 20 μL reactions containing the following: 0.8 μL of template genomic DNA (5–300 ng), 1.25× Buffer B (New England Biolabs), 3.125 mM MgCl$_2$ (New England Biolabs), 0.313 mM dNTPs (New England Biolabs), 1 μM each of forward and reverse primers (Integrated DNA Technologies, Coralville, IA, USA), and 0.2 μL of crude *Taq* polymerase. Reverse primers were 5' labeled with the fluorophore 6-FAM. PCR was conducted in a T100 Thermal Cycler (Bio-Rad Laboratories, Hercules, CA, USA) with an initial denaturation of 94°C for 30 s; 35 cycles of denaturing at 94°C for 30 s, 30 s at the optimized annealing temperature (S1 Table), and extension at 72°C for 30 s; ending with a final extension of 72°C for 3 min. We increased the annealing temperature as the genotyping proceeded in order to reduce background amplification and produce highly-repeatable, easily discernable. Fragments were separated on an ABI 3730xl DNA Analyzer (Cornell Core Laboratories, Ithaca, NY) with a GeneScan LIZ 500 size standard (Life Technologies, Carlsbad, CA, USA). Fragment sizes were determined using PeakScanner Software v1.0 (Life Technologies) using the default settings. Each fragment was manually scored and binned based on peak intensity and fragment length consistent with expected nucleotide tandem repeat sizes. Peaks less than one fifth the size of the tallest peak for each PCR reaction were attributed to stutter and discarded.

To confirm the reliability of our microsatellite data (hereafter referred to as "fragments" because they were treated as dominant markers), we examined the inheritance of all scored fragments by genotyping five controlled crosses for all four loci. Crosses were chosen to maximize the number of segregating markers across all four loci. On average, 12.7 F1 offspring per locus were genotyped to determine whether markers were inherited in a predictable fashion (n = 1–4 offspring per cross). Of 24 fragments initially detected, 16 were examined in the F1 generation. Four fragments that appeared in offspring that were not present in the parents were removed from all subsequent microsatellite analyses.

For most nuclear microsatellite loci and samples, more than two fragments were produced due to the hexaploid nature of ERTE. We were unable to confidently assign dosage and therefore not able to identify individual genotypes. Instead, we treated each marker variant as present or absent and analyzed the data with the restriction site model in Structure v2.3.3 [44] as is often done with microsatellites in polyploids [45]. We tested a range of genetic groupings (k = 1–9) using location priors and allowing for admixture. Runs were first conducted using all identified marker variants including samples from the closely related wallflower ERCAAN to ensure the model could differentiate these taxa (ngen = $10^6$, 4 replicates per k-value, burnin = $5 \times 10^5$, lambda = 2.1237, determined empirically, using eight identified ERTE populations and species distinctions as priors). After confirming the Bayesian clustering analysis could differentiate ERTE from ERCAAN, we ran a comparable analysis with only the ERTE samples using populations as location priors (ngen = $10^6$, 20 replicates per k-value, burnin = $5 \times 10^5$, lambda = 0.3805, determined empirically). The number of genetic groupings that best fit the data was calculated using the Δk method [46] in Structure Harvester [47]. An individual was considered admixed if it exhibited less than 95% assignment probability to a single group.

The distribution of genetic diversity was assessed with an analysis of molecular variance (AMOVA) and population genetic subdivision ($F_{ST}$) in Arlequin v3.5 [48]. For AMOVA, we used the restriction site model and compared the partitioning of genetic diversity within and among populations as well as among the genetic clusters identified by Structure [49]. For genetic subdivision, we calculated pairwise $F_{ST}$ values between all eight ERTE populations. After identifying unique genetic groupings concentrated at the periphery of the species range in the Structure k = 6 analysis, we made *post hoc* comparisons of $F_{ST}$ within the central cluster of populations consisting of highly admixed individuals (AZA, GEY, MTH and OLY) versus between remaining four peripheral populations and the central cluster of populations. $F_{ST}$ values that were significantly different from zero were determined with $10^4$ permutations in Arlequin [48]. We accounted for multiple testing using the most conservative form of the Bonferroni correction.

In order to examine the effects of population size on genetic diversity, population censuses were conducted in 2011–2013 and compared with allelic richness per individual to account for the variable sample sizes across populations (Table 1). Due to the limited number of populations sampled, we applied the non-parametric Spearman rank correlation coefficient to determine significance.

For the isolation by distance analysis, geographic distances were determined in Google Earth based on GPS coordinates of the center of each population. Genetic distances were estimated using pairwise $F_{ST}$ with the scaled metric $F_{ST} / (1 - F_{ST})$ [50]. To compare the geographic distance matrix with the pairwise $F_{ST}$ distance matrix, we used the nonparametric Mantel test with $10^4$ permutations in PASSaGE v2.0 [51].

## Results

### Genome sizing

The mean genome size in leaf tissue from ERTE is 2C = 2.92 pg (2.82–3.06). This translates to approximately 2.86 billion base pairs per nucleus.

### Genome skimming

Pooled individuals from eight populations were barcoded and sequenced on a single lane of Illumina HiSeq2000 producing an average of 624,277 x 250 bp paired-end reads per sample (range 481,796–722,138). Low quality ends of an average of 272,242 reads per sample were trimmed before assembling (Raw data available from GenBank's Short Read Archive Accession

# SRR10356050-SRR10356059). Separate reference guided assemblies of the chloroplast genome, nrDNA and mitochondrial genome were 95.9%, 100% and 28.4% complete, respectively. Subsequent alignments of the nearly complete chloroplast genome and complete nrDNA cistron were relatively straightforward (see dedicated subsections below). In contrast, assembly of the mitochondrial genome revealed very low coverage (mean = 3.95x). Attempts at aligning these partial mitochondrial genome assemblies were severely impaired by the large amounts of missing data (mean = 71.6%). The small portions that were aligned contained numerous ambiguities (mean = 4.24%) or were invariant because they represented coding regions of housekeeping genes. Due to the low coverage, large amount of missing data, frequent ambiguities and lack of variation, the mitochondrial genome was not used in subsequent analyses.

**Analysis of the chloroplast genome alignment.** The nearly complete chloroplast genome alignment including the two ERCAAN populations measured 154,453 bp with 32,130 variable sites (considering ambiguities) and 154,432 bp with 25,289 variable sites for just the eight ERTE populations (16.38% variability) (Genbank Accession numbers MN626590-MN626599). There was an average of 6.55% of missing bases per sample (range from 5.26 to 10.3%). An average of 24,938 reads mapped to the *A. thaliana* chloroplast genome reference per sample (average coverage per base pair = 39.8x, range from 18.6x to 67x). Maximum likelihood phylogenetic analysis using RAxML produced a tree with five well-supported branches (Fig 2A). ERTE samples formed a strongly supported clade to the exclusion of the two ERCAAN samples (BS = 100%). Within ERTE, there were two strongly supported lineages: GEY, OLY, QH, SHGW and MTH (BS = 100%) versus AZA, BD and HWY (BS = 100%). Within the former clade, OLY, QH and SHGW segregate from GEY and MTH with moderate bootstrap support (BS = 79%).

DAPC analysis of the chloroplast genome based on two partitions (ERCAAN vs. ERTE) unequivocally assigned membership to the two distinct species for all populations sampled (probability of membership = 100%; S2A Fig). Likewise, the probability of membership was unequivocal when the analysis was expanded to three groupings (ERCAAN plus the two distinct lineages recovered in the phylogenetic analysis of the chloroplast genome = AZA, BD and HWY vs. GEY, OLY, QH, SHGW and MTH). The probability of membership of the samples to these three groupings is 100% (S2B Fig). These three groups were clearly separated along the first two principle component axes of the DAPC (representing 73.6% of the variation), showing no overlap of their variation measured by their 95% inertial ellipses (Fig 3).

**Analysis of the nrDNA alignment.** The complete nrDNA alignment along with the two ERCAAN populations (including 18S, ITS1, 5.8S, ITS2, 28S) measured 5,817 bp with 59 variable sites (considering ambiguities) and 33 variable sites among the ERTE samples alone (0.57%) (Genbank Accession numbers MN622144-MN622153). An average of 7,553.3 reads per sample mapped to the mosaic reference (average coverage per base pair = 191.7x, range from 109.3x to 289.7x). Maximum likelihood phylogenetic analysis using RAxML produced a tree with only a single well-supported branch defining the monophyly of the ERTE samples as separate from the two ERCAAN samples (BS = 100%; Fig 2B).

Since all but one of the 59 SNPs found in the nrDNA alignment have an ambiguity for at least one sample, and DAPC does not treat ambiguities as both underlying bases (instead treats them as a separate character or ambiguities can be ignored depending on user defined settings following "SNP<-DNAbin2genind"), we were unable to conduct the DAPC analysis for the nrDNA.

## Microsatellite variation

Null alleles can complicate microsatellite analyses [52,53]. One symptom of null alleles is a large number of failed PCR reactions for a single locus [54]. Out of 699 PCR reactions, only

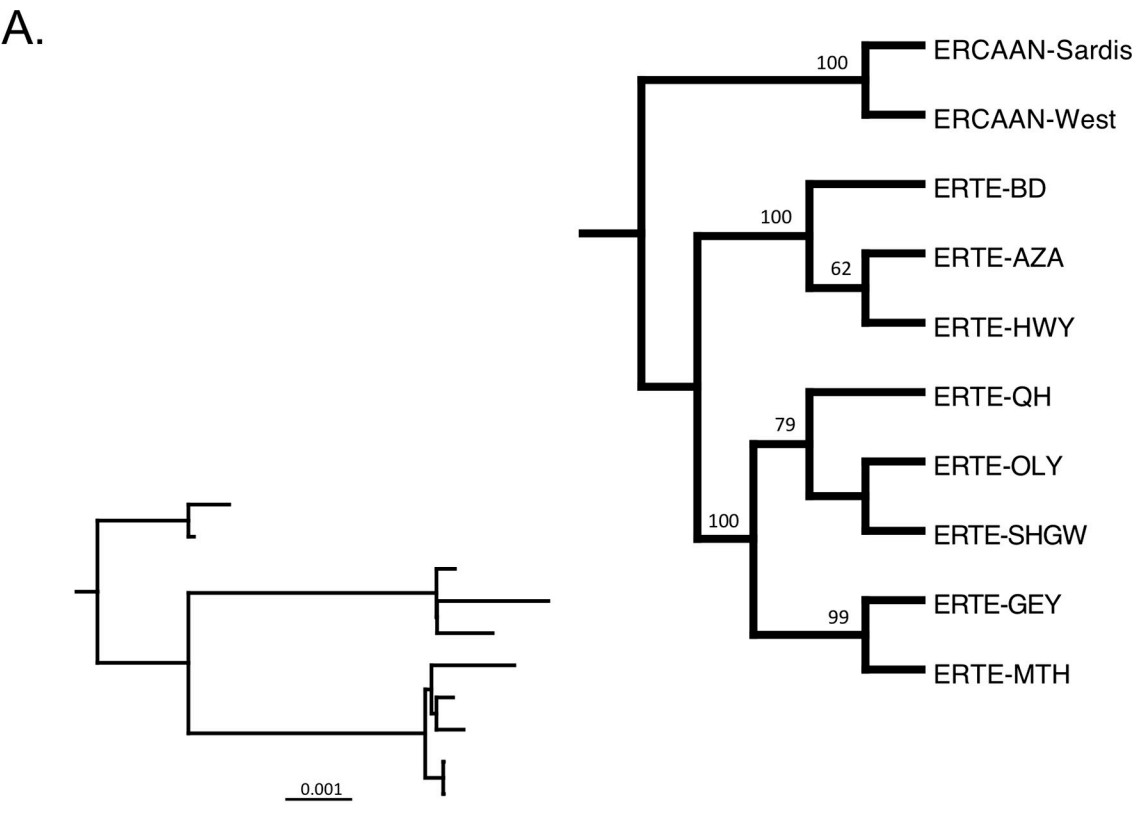

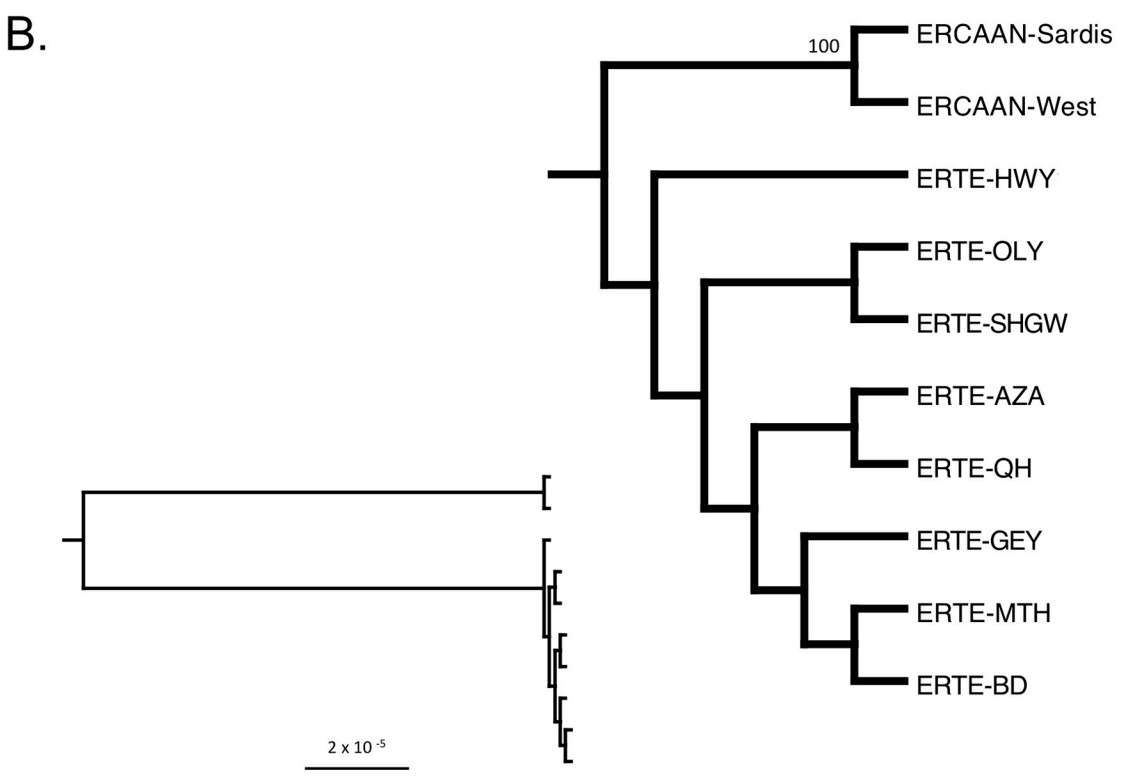

**Fig 2. Maximum likelihood phylogenetic results for the eight populations of *Erysimum teretifolium*.** Phylogenetic relationships were determined from the nearly complete chloroplast genomes (A) and complete nuclear ribosomal cistron (B). The trees are rooted with two closely related ERCAAN samples and analyzed with RAxML. Bootstrap values were shown above branches (only BS > 50% are displayed). The larger cladograms show relationships among populations while branch lengths are displayed in the inset phylograms with populations in the same vertical order. For the phylograms, branches are drawn proportional to the number of substitutions per site (see scale bar). Population abbreviations are defined in Table 1.

two samples failed to amplify and for separate loci (D10 in one case and D4 in the other)–however, this is just one method of identifying null alleles for nuclear microsatellites (see [54] for a comprehensive review). Twenty-four nuclear microsatellite fragments were initially identified (2–9 per locus; S1 Table). After genotyping our controlled crosses, we identified four fragments that appeared in the F1 offspring but were not present in the parents (S1A–S1O Fig). These were removed from all subsequent analyses. Populations harbored an average total of 13.9 fragments for all four loci (Table 1). Eleven fragments were present in all eight populations. Of these eleven fragments, three were fixed in all populations, and four were fixed in at least one population.

**Bayesian genetic clustering.** To determine the reliability of the restriction model in accurately identifying genetic clusters and assigning individuals, we ran a Structure analysis with ERTE and the morphologically and geographically distinct ERCAAN. In this analysis, k = 2 was the preferred number of genetic clusters (Fig 4A). Assuming an individual is admixed if it has < 0.95 assignment to a single genetic grouping, 99.57% of ERTE individuals were assigned confidently to one group with no admixture. Meanwhile, ERCAAN individuals were largely

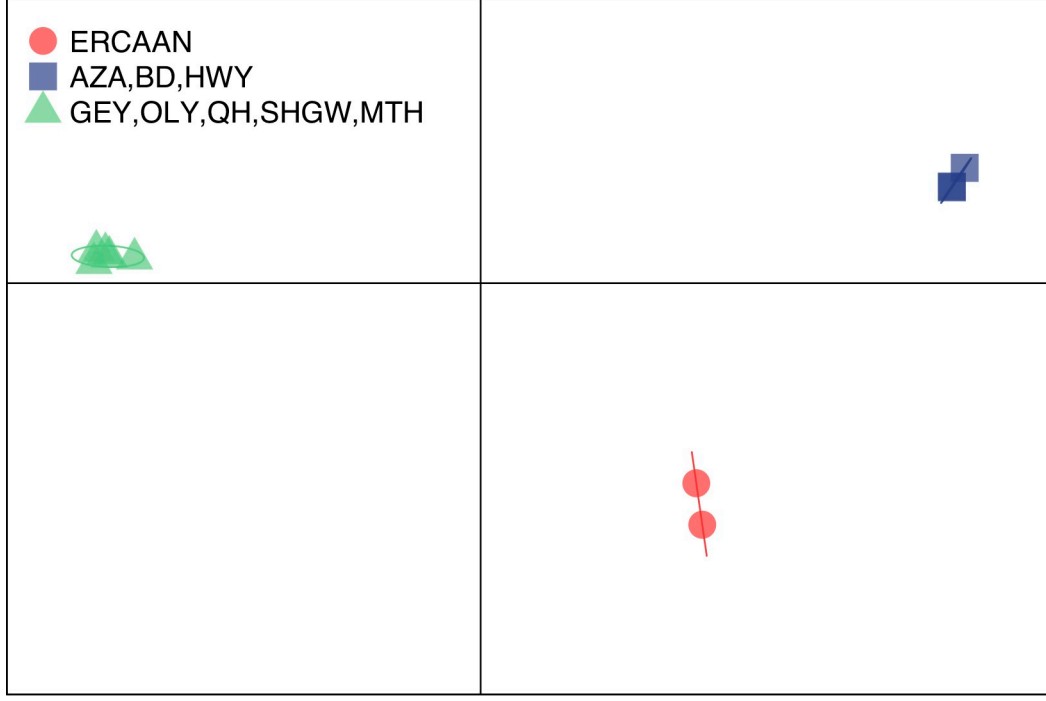

**Fig 3. Scatter plot of a discriminant analysis of principal components (DAPC) of the chloroplast genome.** The *a priori* groups based on the two distinct lineages recovered in the phylogenetic analysis of the chloroplast genomes are represented by different symbols and colors (ERCAAN populations = red circles; AZA, BD and HWY populations = blue squares; GEY, OLY, QH, SHGW and MTH populations = green triangles). The 95% inertial ellipses around each cluster represent the variance of the two first principal components of the DAPC analysis. Population abbreviations are defined in Table 1.

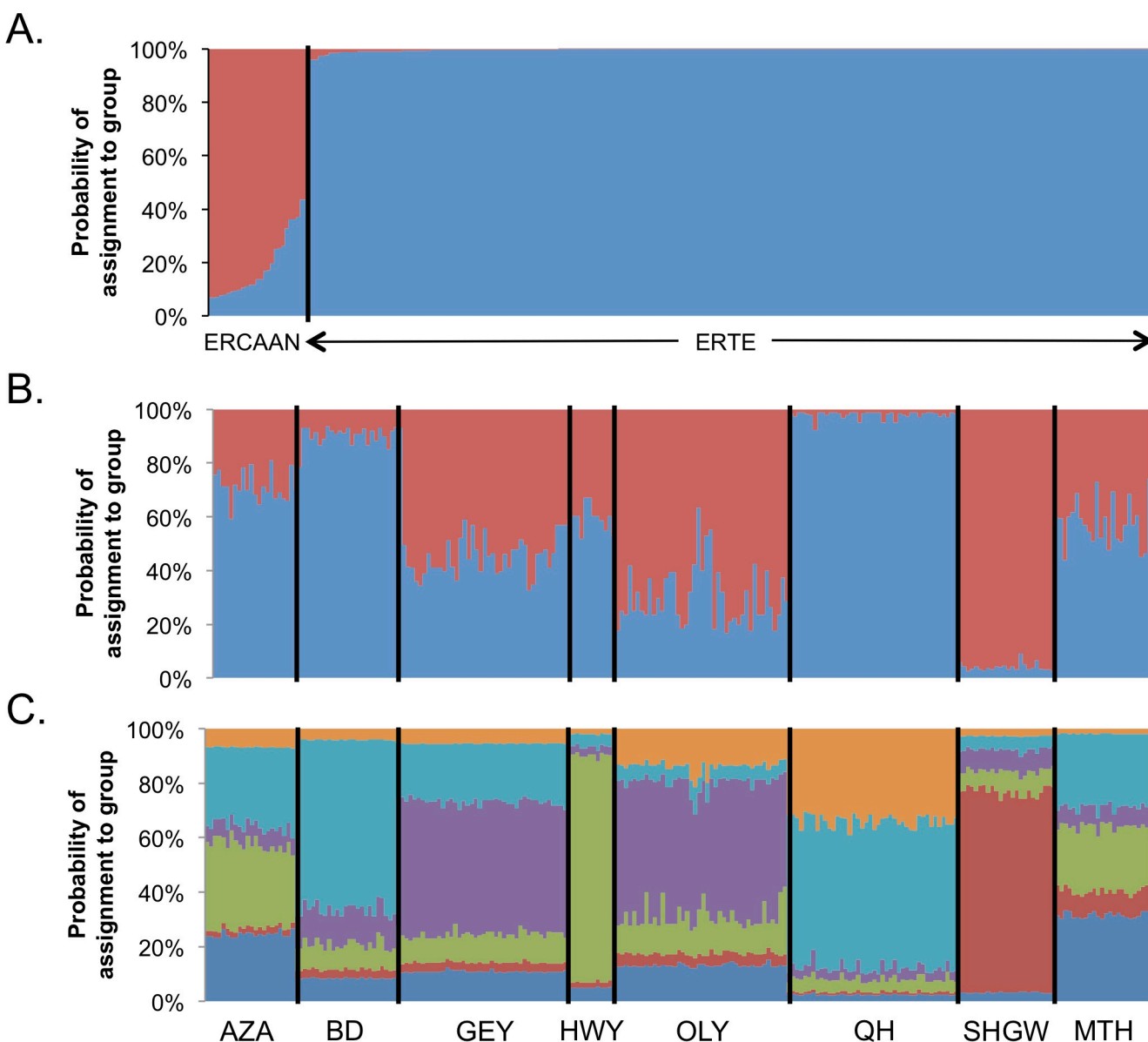

**Fig 4. Structure results for ERTE and ERCAAN with various subdivisions.** Assignment probabilities for individuals sampled from eight ERTE populations and two ERCAAN populations for the optimal k = 2 (A) and similar analyses for just the ERTE samples for k = 2 (B) and k = 6 (C). Population abbreviations are the same as in Table 1.

admixed, but mostly assigned to a distinct genetic grouping (mean proportional assignment to the unique grouping was 0.812). From this, it appears the restriction model is capable of detecting genetic subdivision in our heterospecific nuclear microsatellite data, so we proceeded with more detailed Structure analyses with just the ERTE samples.

We performed separate Structure analyses for the eight ERTE populations spanning one to nine genetic groupings–representing one more than the number of populations sampled. The most likely number of genetic groupings was k = 6, followed by k = 2 (Fig 4, S3 Fig). Assuming an individual is admixed if it has <0.95 assignment to a single genetic grouping, the k = 6 (Fig 4C) analyses revealed 100% admixture for all individuals in all populations. The k = 2 analysis

**Table 2. AMOVA results from microsatellite analysis.**

| Source of Variation | Degrees of Freedom | Sum of Squares | Variance Components | Percentage of Variation |
|---|---|---|---|---|
| Among groups | 3 | 34.8 | 0.18 | 12.0 |
| Among populations | 4 | 15.5 | 0.08 | 2.66 |
| Within populations | 225 | 303.7 | 1.35 | 85.3 |
| Total | 232 | 353.9 | 1.61 | |

revealed 73.82% of individuals were admixed, with QH and SHGW individuals having the largest percent of individuals without admixture (2.38% and 16.67%, respectively; Fig 4B).

## Partitioning of genetic diversity and isolation by distance

To determine the distribution of genetic diversity within and among ERTE populations, we used AMOVA and $F_{ST}$. The AMOVA estimated within population, among population and among grouping (using the Structure k = 6 groupings in Fig 4C). The majority of molecular variation exists within populations (85.3%) (Table 2). Only 2.66% of genetic variation exists among populations, whereas 12.0% of genetic variation is explained by the Structure-identified groupings.

For genetic subdivision, pairwise $F_{ST}$ averaged 0.153 (0.008–0.431) with 18 out of the 28 comparisons being statistically significant even after applying the most conservative Bonferroni correction with no correlation ($p < 0.0018$; Table 3). All $F_{ST}$ comparisons involving SHGW or HWY were significantly different from zero (mean $F_{ST}$ = 0.217 and 0.294, respectively). $F_{ST}$ within the central cluster of populations (AZA, GEY, MTH, and OLY) averaged 0.055 (0.008–0.106). In comparison, $F_{ST}$ between peripheral populations (BD, HWY, QH, and SHGW) and the central cluster of populations averaged 0.200 (0.034–0.431).

Population census sizes range from 35–1000. Allelic richness measured as the average number of fragments per sampled individual ranged from 9.04–10.21. There was no significant correlation between population census size and allelic richness per sampled individual (Fig 5; Spearman rank correlation, $r_s$ = -0.47619, df = 6, p = 0.233.).

Pairwise geographic distances between populations averaged 4900 m (range: 480–12440 m). Isolation by distance for all population comparisons was not significant (Fig 6; Mantel test, t = 1.142, p > 0.05). Since the most geographically disjunct population (BD) may have received seeds from QH during unsanctioned restoration efforts, we also examined isolation by distance without BD. The pattern of isolation by distance is strongly significant without BD (Fig 6, $r^2$ = 0.561; Mantel test, t = 2.232, p = 0.005). After excluding the BD comparisons, the scaled genetic subdivision metric, $F_{ST} / (1 - F_{ST})$, increased by 0.07 per kilometer.

**Table 3. Pairwise $F_{ST}$ values between all populations.**

| Population | AZA | BD | GEY | HWY | OLY | QH | SHGW |
|---|---|---|---|---|---|---|---|
| BD | 0.034 | | | | | | |
| GEY | 0.051* | 0.051* | | | | | |
| HWY | 0.153** | 0.386** | 0.287** | | | | |
| OLY | 0.073* | 0.127** | 0.025* | 0.291** | | | |
| QH | 0.072* | 0.048* | 0.123** | 0.327** | 0.168** | | |
| SHGW | 0.187** | 0.211** | 0.119** | 0.431** | 0.113** | 0.260** | |
| MTH | 0.008 | 0.080* | 0.059* | 0.183** | 0.106** | 0.103** | 0.195** |

* $p < 0.05$

** $p < 0.0018$ (following full Bonferroni correction)

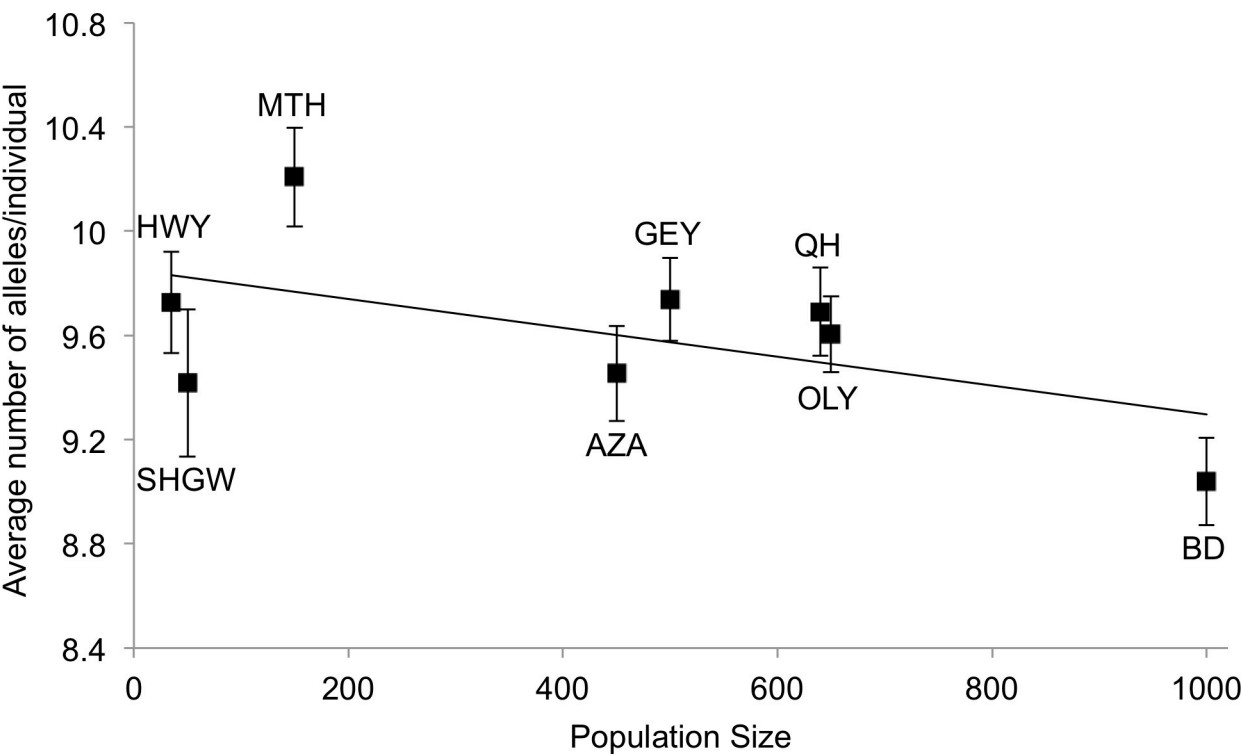

**Fig 5. Population size and allelic richness.** The average number of alleles per sampled individual for the eight ERTE populations is not significantly correlated with population census size (Spearman rank correlation, $r_s$ = -0.45238, p > 0.05). Error bars represent the standard error of the mean for allelic richness. Populations are abbreviated as in Table 1.

## Discussion

Genome skimming yielded a nearly complete chloroplast genome that was deeply divided among the *E. teretifolium* populations sampled, yet very little variation was detected in the nuclear ribosomal cistron. In contrast, the nuclear microsatellite analysis indicated the majority of genetic variation was found within populations with limited (yet significant) population differentiation.

### Genome skimming produced complete plastid genome and nuclear ribosomal cistron

Genome skimming successfully generated the chloroplast genome and nuclear ribosomal cistron, but was unable to recover the mitochondrial genome. For the chloroplast genome, we had to mask regions of low coverage and regions with excessive ambiguities. These ambiguities could have arisen from two sources: (1) small regions of low sequencing coverage, and/or (2) genetic variation within the pooled individuals. The relatively high average coverage in the chloroplast genome sequencing makes technical errors less likely and a visual inspection suggests this was not the primary cause. Alternatively, the existence of numerous shared, phylogenetically informative ambiguities in the chloroplast genome alignment suggests substantial within population variation (consistent with the microsatellite results showing most variation within populations). Unfortunately, we did not have sufficient coverage to deconvolute these ambiguities into allele frequencies. Therefore, we treated them as ambiguities, which were recognized in the phylogenetic analysis, yet ignored in the DAPC analysis.

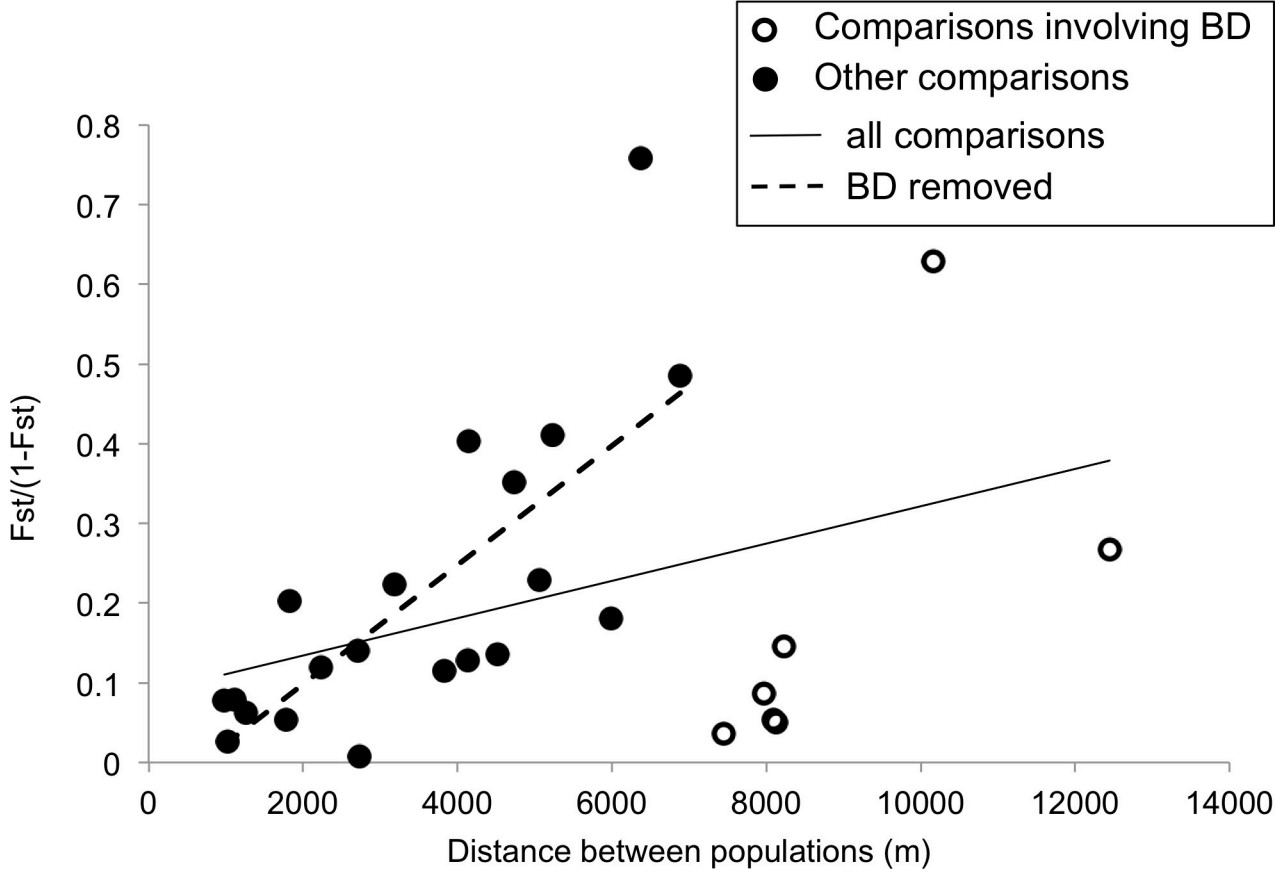

**Fig 6. Isolation by distance.** The correlation between genetic subdivision (scaled $F_{ST}$) and geographic distance is not significant when all populations are considered (solid line; Mantel test, $p > 0.05$). The broken line depicts a significant correlation after removing the BD comparisons indicated with open circles (Mantel test, $p < 0.005$).

Similarly, the ribosomal cistron was completely sequenced with very high coverage, but had very little variation and largely homoplasious. Here we also saw cases of shared ambiguities among populations even though we had excessively high coverage for the entire nrDNA. For the same reason stated above, we chose to treat the each nrDNA pooled sample as a population consensus rather than try to disentangle the proportion of reads at ambiguous sites into SNP frequencies at each site.

Conversely, we were unable to reliably assemble the mitochondrial genome despite the abundance of this organelle in genomic DNA. Since plant mitochondria undergo frequent structural rearrangements and remarkably variable substitution rates within and among species [55–58], genome skimming is unlikely to fully recover the complete mitochondrial genome as reported by others [32,59,60]. For example, mitochondrial assemblies in *Silene* were largely incomplete and mainly restricted to coding regions, contrasting with the assembly success of chloroplast genome and nrDNA [61]. In addition, the phylogenetic relationships based on the mitochondrial genome typically contain less phylogenetic signal that those based on the chloroplast genome [32,61].

The phylogenetic analyses of the chloroplast genome confirm the genetic uniqueness of ERCAAN vs. ERTE and suggest a strong subdivision within ERTE. The split of BD, HWY and AZA from the remaining populations does not match biogeographic predictions, nor any known morphological or ecological differences. However, the association of GEY and MTH

(BS = 99%) and QH, OLY and SHGW (BS = 79%) partially reflect geographic proximity (Fig 1B). This largely enigmatic pattern of chloroplast genome diversity on the landscape could be caused by the arbitrary fixation of ancestral polymorphisms or recent introgression (through seed dispersal) [61]. Alternatively, inaccurate chloroplast genome assembly could explain this unexpected genetic subdivision within ERTE. However, in a previous study we validated the veracity of next-generation data by Sanger sequencing specific region of three genomes with a high number of variable sites [61], indicating that assembly error is unlikely to have caused these results. Although the cpDNA tree topology appears incongruent with that of the nrDNA region, the latter lacks any branches within ERTE with bootstrap values > 50%. However, the cpDNA phylogenetic results are inconsistent with the microsatellite results, which show most of the genetic variation within and not among populations–most likely due to the difference between maternally-inherited organellar genome vs. homogenized nuclear genome by concerted evolution within individual genomes and/or among individuals within a population [62], resulting in distinct evolutionary histories [61,63–65].

## Microsatellite analysis

Microsatellites are powerful markers for estimating genetic variation within and among populations, yet only when applied correctly. Herein, we used heterologous microsatellites designed from a closely related species (*E. mediohispanicum*) which can lead to increased chances of null alleles [54], yet null alleles appear to be rare since only 2 of our 699 PCR reactions failed repeatedly (0.29%). However, our power to detect null alleles was limited to PCR reactions that produced no fragments–we would not have detected missing fragments as long as one allele amplified. Mutations at or near the priming sites linked with particular alleles can reduce the number of allele variants in a heterozygote, which can be particularly difficult to detect in polyploids [54].

Twenty of our 24 markers (83.3%) showed the expected pattern of inheritance in controlled crosses indicating most markers are behaving as expected. Because of conservation concerns at four populations (BD, GEY, QH and OLY), we included individuals grown from seed collected from distinct maternal families to avoid impacting the natural population. If we found any unusual patterns of genetic diversity in these populations, we may have attributed it to not representing "standing variation" since these individuals were never vetted by the filter of natural selection in the wild. However this was not the case and therefore the use of offspring from maternal families in a few cases was unlikely to have introduced any unexpected patterns of variation.

The most likely limiting factor in our microsatellite analysis is having to treat these markers as present/absent because of the hexaploid nature of ERTE (and having to apply the restriction model for subsequent analyses–see [45] for comparable approaches to treating microsatellites as dominant markers in polyploids). Where the restriction model and our genetic sampling may have been most limiting (Structure analysis), we employed a positive control by including ERCAAN samples. Structure was able to confidently distinguish ERCAAN from ERTE suggesting our treatment of the microsatellites was sufficient to detect any deep population subdivision. Fine scale genetic partitioning within ERTE is probably not possible with this dataset as evidenced by the substantial admixture. Therefore, we have emphasized the within and among population results of the AMOVA and $F_{ST}$ calculations that should be more robust to the restriction model and our genetic sampling.

## Mating system predicts the distribution of most microsatellite genetic diversity

Species with outcrossing mating systems are predicted to harbor most of their genetic variation within populations and exhibit very little population subdivision [66,67]. Alternatively, species

that occupy island-like habitats surrounded by barriers to gene flow should have substantial population subdivision with larger amounts of genetic diversity attributed to comparisons among populations [4]. Our microsatellite analysis indicates over 85% of genetic variation is housed within populations, consistent with the self-incompatible mating system in ERTE [27]. Only 2.66% of the genetic variation was attributed to among population comparisons, contrary to the prediction of sandhill species being isolated like a true island endemic. The dominant effect of the self-incompatible mating system in ERTE has also been reported from Australian dioecious *Atriplex nummularia*, where 87.52% of microsatellite genetic variation was found within populations [68], yet this species does not reside on islands, nor does it occupy an island-like habitat. In reviewing the distribution of genetic diversity in 263 plant species, Duminil et al. [18] found that "mating system is the main influencing factor on $F_{ST}$" due to increased pollen flow and decreased genetic drift in outcrossers. Interestingly, even in island plant species, mating system consistently drives the distribution of genetic diversity [4]. In 10 out of 11 studies of island plant population genetics reporting AMOVA results, the mean percent genetic variation attributed to among population comparisons was low (38.72%), yet ERTE is substantially lower than this (2.66%). The lowest among-population AMOVA result was from an AFLP study of an island species of fir tree from Taiwan (*Cunninghamia konishii*) where only 12.21% of genetic variation was explained by comparisons among populations. The remaining 87.79% of genetic variation was attributed to within population comparisons likely due to the effectiveness of wind-pollination in overcoming the barriers among populations [69]. Similarly, in the putatively outcrossing *Weigela coraeensis* from the Inzu Islands and Japanese mainland, 76.21% and 76.07% of genetic variation was attributed to within population comparisons for allozymes and microsatellites, respectively [70]. Clearly, mating system drives the distribution of most genetic diversity, even in island and island-like habitats.

Although the self-incompatible mating system may be responsible for the maintenance of high levels of genetic variation within populations because of its positive effect on $N_e$, gene flow among populations relies on either pollinators or seed dispersal. ERTE attracts a diversity of pollinators from Hymenoptera, Lepidoptera, and Diptera [27]. Lepidopterans such as the Chalcedon Checkerspot (*Euphydryas chalcedona*) could seemingly overcome the relatively short distances and habitat barriers between populations to encourage gene flow [27]. Several species of small and large solitary bees as well as numerous bumblebees (*Bombus*) have been observed pollinating ERTE [27], most of which could easily travel distances of several kilometers [71,72].

Although less likely, seed dispersal represents another vector of gene flow that could contribute to the observed distribution of genetic diversity. ERTE does not have any obvious fruit or seed dispersal mechanisms, anemochory could be responsible for transferring occasional seeds between populations since wind moves seeds across oceans in other taxa [73]. However, the predominant wind direction across the species range blows from the northwest. This would facilitate gene flow from the central populations to the two small, peripheral populations HWY and SHGW. Yet, these two populations are the most genetically distinct, suggesting the presence of significant barriers to gene flow. In addition, it appears that most seeds and seedlings are concentrated near the base of maternal plants. Long-distance seed dispersal in ERTE is likely rare and difficult to quantify, yet it could have a large effect on the distribution of genetic diversity [74].

## Island-like patterns explain the residual microsatellite genetic diversity

Although the vast majority of genetic diversity in ERTE resides within populations, the partitioning of the residual genetic diversity reflects some aspects of insular lineages. Island species

are predicted to (1) show genetic subdivision among islands that is (2) positively related to the separation distance [4]. In ERTE, we found significant pairwise $F_{ST}$ values in 64% of comparisons between these island-like sandhill populations (although the $F_{ST}$ values were low, they were often significantly different from zero). This includes all comparisons involving the two smallest, peripheral populations (HWY and SHGW). There is a trend towards isolation by distance that becomes significant after removing the geographically distant, yet not very genetically distinct BD population, potentially due to unofficial trafficking of seeds from QH to BD (J. McGraw & V. Haley, personal communication), however BD and QH fall into separate, strongly supported cpDNA lineages (Fig 2A). In the isolation by distance analysis for the microsatellites, BD displays less genetic divergence than predicted by its substantial geographic distance (over 7 km to the nearest population; Fig 1B). Structure analysis of the microsatellite data (k = 2) suggests BD and QH are very similar genetically–most individuals from both populations have the same proportions of admixture (Fig 4B). In a modified isolation by distance analysis, we assigned BD pairwise geographic distances as if it originated from QH and found the genetic distance from BD to other populations fits the regression as if it came from QH ($r^2$ = 0.41, p < 0.05).

Although it is common to find significant population subdivision among islands [8,70], detecting isolation by distance is considerably less common [4]. Only one of seven studies investigating the genetics of island plants showed significant isolation by distance [4]. This single study showing isolation by distance focuses on a dune stabilizing grass from the southeastern Atlantic and Gulf coasts of the United States–a species that is only occasionally on near shore islands [75]. In a meta-analysis of 240 isolation by distance data sets, including animals and plants, this simple model explains a relatively small proportion of variance in genetic structure in most studies (mean $r^2$ = 0.22) [76].

The distribution of genetic diversity in ERTE is consistent with the abundant center model [12]. Population-level Structure analysis suggests the presence of a core cluster of populations that are composed largely of admixed individuals with low probability of assignment to any single genetic grouping. It is unlikely that this is an artifact of applying the restriction model, nor to lack of signal in the data since our control Structure analyses clearly distinguished the very closely related ERCAAN from ERTE (Fig 4A). Furthermore, peripheral populations have an average pairwise $F_{ST}$ value that is 3.63× that of the populations from the central cluster. The central cluster (AZA, GEY, MTH, and OLY) has the least significant barriers to gene flow and may have been historically more connected prior to sand mining and residential development [24].

There is no relationship between allelic richness and population size–a proxy for island area in the MacArthur-Wilson model of island biogeography [77]. Counter-intuitively, the trend is towards decreasing allelic richness as population size increases, yet the lack of variation in allelic richness among the eight populations clearly undermines any possible correlation with population census size (Fig 5). On true oceanic islands, there is often no correlation between genetic diversity and population size [13], nor between genetic diversity and island size [8,70,78]. Yet, in a review of genetic diversity and population size, 22 of 23 allozyme studies of plants and animals had significant correlations between population size and genetic diversity [79]. Although the measurement of genetic diversity is relatively straightforward, quantifying a meaningful population size that reflects the present and historical number of effective individuals in a population is much more complicated [80,81]. In this case, it appears that the self-incompatible mating system has maintained a relatively constant average allelic richness per population irrespective of the contemporary census size. Alternative measures of genetic variation, such as heterozygosity, may be better suited for examining a correlation with population census size, but we were unable to assign genotypes because of ERTE's hexaploid genome.

## Genome sizing and microsatellites in polyploids

We confirmed that ERTE is a hexaploid by estimating its genome size using flow cytometry. Our estimates (2C = 2.92 pg) are 2–3 × larger than genome sizes estimated for diploid Eurasian *Erysimum* species with smaller chromosome counts (*E. scoparium* 2n = 28 & 2C = 1.08 pg; *E. bicolor* 2n = 28 & 2C = 1.16 pg; *E. chieranthoides* 2n = 16; 2C = 1.66 pg; http://data.kew.org/cvalues/; [82]) consistent with a tripling of the diploid genome size followed by some post-polypoidy genome size contraction [83]. This represents the first genome size estimation from the *E. capitatum* alliance [29]. The lack of variation in genome size among ERTE populations suggests there is little, if any polyploidy within this species. The polyploidy event underlying the large genome size in ERTE likely predated the radiation of the *E. capitatum* alliance since all 25 taxa have hexaploid chromosome counts [29].

These results are consistent with observations of up to six alleles per locus in ERTE, ERCAAN and other members of the *E. captiatum* alliance. Others have treated microsatellite data in polyploids with several different approaches including trying to discern genotypes using the dosage of each fragment, making HWE assumptions to estimate heterozygosity [84], using a modified measure of heterozygosity ($H'_e$ and $H'_o$) and genetic subdivision ($F'_{ST}$) that emphasizes the presence of shared bands [85], and simply treating the fragments as present/absent, then applying the restriction model in Structure and Arlequin [44,48]. After confirming the inheritance of our microsatellite fragments by genotyping controlled crosses, we showed that Structure's restriction model could differentiate ERTE from the closely related subspecies, ERCAAN. Genotyping controlled crosses identified four out of 24 originally scored fragments that appeared in offspring, but were not present in parents–a critical step in scoring and analyzing microsatellites in a polyploid such as ERTE.

## Conservation implications

The distribution of genetic diversity described herein can be used to guide future conservation efforts and help determine appropriate seed sources for reintroduction efforts. Given that more than 80% of all genetic variation based on nuclear microsatellites is found within any individual population, one could preserve the vast majority of genetic variation in a single, large population. Yet, in order to capture the remaining genetic variation, peripheral populations (such as QH, HWY, and SHGW) must also be preserved. Preservation of these strongly differentiated, marginal populations should receive high priority [86]. In particular, HWY and SHGW are not only the smallest and most threatened populations, but neither are currently being managed to maintain the sandhill parkland habitat critical for ERTE [24]. Additionally, some of these peripheral populations harbor the genetic uniqueness of the two distinct lineages found for the chloroplast genome (such as HWY and SHGW). The conservation of these marginal populations would not only preserve the largest proportion of nuclear genetic variation of ERTE populations, but would also capture the genetic diversity in the chloroplast genome. In addition to preserving neutral genetic variation like that measured here, we strongly recommend measuring and conserving the maximum number of self-incompatibility alleles to prevent future reproductive failure like that seen at BD and in other taxa [87]. Maintaining large effective population sizes and a healthy pollinator fauna are critical in managing the existing ERTE populations.

Bonny Doon has recently experienced a precipitous population decline, from over 2,000 individuals in 1994 to only six individuals in 2013 [30]. The genetic similarity between BD and QH corroborates reports that seeds may have been moved from QH to BD in attempts to "rescue" that declining population (however, the cpDNA comparisons place QH and BD is separate, well-supported clades). Therefore, it is unclear if any unique gene pool that may have

existed at BD may already have been diluted. Translocating individuals within a species range to augment a declining population is at least a viable short-term measure to save at-risk populations [88] and may be viable in other cases when overall population differentiation is low, like in ERTE. Additionally, mixing more source populations may reduce the possibility of limited genetic diversity (especially in regard to self-incompatibility alleles) via the founder effect when initial reintroduction population sizes are small [89]. Complementary studies of inbreeding and outbreeding depression and local adaptation to BD edaphic conditions are currently underway and must be used in coordination with these genetic results when choosing source material for a reintroduction [27].

## Conclusions

The majority of genetic diversity measured from nuclear microsatellites in the self-incompatible, sandhill endemic ERTE resides within populations, consistent with its obligate outcrossing mating system. The small amount of residual variation fits some predictions of island species (lack of gene flow among island-like populations and weak isolation by distance), but not others (no effect of population size on allelic richness). Based on these results, mating system has a substantially larger effect on the distribution of genetic variation than the island-like sandhill habitat. Some populations at the margins of the species range are genetically differentiated and their preservation should be prioritized accordingly. By preserving these populations, we also capture both of the two distinct lineages found in the chloroplast genome, increasing the value of such conservation efforts. For reintroduction efforts at BD, a thorough sampling of any individual population would capture the vast majority of microsatellite genetic variation, yet seeds from multiple sources including the smaller, peripheral populations like HWY and SHGW should be included to maximize the amount of genetic diversity, especially in relation to self-incompatibility alleles. Future conservation efforts on islands and island-like habitats must consider biological factors controlling the distribution of genetic diversity in addition to their unique insular habitats.

## Supporting information

**S1 Fig. Inheritance of microsatellite fragments.** Parental and F1 chromatograms for locus D10 (A-E), locus D4/D4b (F-J), and locus C5 (K-O). The four fragments that appear in F1 individuals that are not present in the parents are indicated with asterisks.
(PDF)

**S2 Fig. Barplots based on a discriminant analysis of principal components (DAPC) of the chloroplast genome, in which the group assignment probability of populations is represented.** Two groupings (A) unambiguously differentiate ERCAAN (dark grey) and ERTE (light grey) populations. Three groupings (B) were based on the two distinct lineages recovered in the phylogenetic analyses of the chloroplast genomes (ERCAAN = dark grey; AZA, BD and HWY populations = grey; GEY, OLY, SHGW, MTH and QH populations = light grey). Population abbreviations are defined in Table 1.
(PDF)

**S3 Fig. Structure harvester results for *E. teretifolium* analysis.** Using only samples from the eight populations of *E. teretifolium*, Structure Harvester identified k = 6 as the most likely grouping. Genetic groupings of k = 2 is also considered. Likelihood of K is indicated in A and ΔK is indicated in B.
(PDF)

**S1 Table. Microsatellite fragment diversity, sizes, and optimized annealing temperature.** (DOCX)

## Acknowledgments

The authors are very grateful to "Team Wallflower" and especially Miranda Melen for her invaluable help throughout this study. Dena Grossenbacher and the anonymous reviewers provided helpful comments on an earlier draft. We thank the Land Trust of Santa Cruz County, Jodi McGraw and Terris Kasteen for allowing us to sample several populations under their jurisdictions. Mary-Ann Showers and Cherilyn Burton (United States Fish and Wildlife) were instrumental in encouraging and managing the project. Dr. Ingrid Parker and Jim Velzy (University of California Santa Cruz) provided invaluable guidance early on and shared plants grown from critical populations so we did not have to further impact these endangered species.

## Author Contributions

**Conceptualization:** Justen B. Whittall.

**Formal analysis:** José Carlos del Valle, Julie A. Herman.

**Funding acquisition:** Justen B. Whittall.

**Investigation:** José Carlos del Valle, Julie A. Herman, Justen B. Whittall.

**Methodology:** José Carlos del Valle, Julie A. Herman.

**Supervision:** Justen B. Whittall.

**Writing – original draft:** José Carlos del Valle, Julie A. Herman, Justen B. Whittall.

**Writing – review & editing:** José Carlos del Valle, Julie A. Herman, Justen B. Whittall.

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
