## [Decision Letter · Decision Letter 0]

2 Mar 2020

PONE-D-19-33086

Genome skimming and microsatellite analysis reveal contrasting patterns of genetic diversity in a rare sandhill endemic (Erysimum teretifolium, Brassicaceae)

PLOS ONE

Dear Dr. Whittall,

Thank you for submitting your manuscript to PLOS ONE. After careful consideration, we feel that it has merit but does not fully meet PLOS ONE’s publication criteria as it currently stands. Therefore, we invite you to submit a revised version of the manuscript that addresses the points raised during the review process.

We would appreciate receiving your revised manuscript by Apr 16 2020 11:59PM. To enhance the reproducibility of your results, we recommend that if applicable you deposit your laboratory protocols in protocols.io, where a protocol can be assigned its own identifier (DOI) such that it can be cited independently in the future. For instructions see: http://journals.plos.org/plosone/s/submission-guidelines#loc-laboratory-protocols

We look forward to receiving your revised manuscript.

Kind regards,

Kuo-Hsiang Hung

Academic Editor

PLOS ONE

Journal Requirements:

2. We noted in your submission details that a portion of your manuscript may have been presented or published elsewhere. ["This manuscript is an improved version of a thesis presented by JAH in partial fulfillment of requirements for the SCU Honors Program. “Team Wallflower,” especially Miranda Melen, was instrumental to the completion of this project. Dena Grossenbacher and the reviewers provided helpful comments."] Please clarify whether this publication was peer-reviewed and formally published. If this work was previously peer-reviewed and published, in the cover letter please provide the reason that this work does not constitute dual publication and should be included in the current manuscript.

4. We note that Figure #1 in your submission contains satellite images which may be copyrighted. All PLOS content is published under the Creative Commons Attribution License (CC BY 4.0), which means that the manuscript, images, and Supporting Information files will be freely available online, and any third party is permitted to access, download, copy, distribute, and use these materials in any way, even commercially, with proper attribution. For these reasons, we cannot publish previously copyrighted maps or satellite images created using proprietary data, such as Google software (Google Maps, Street View, and Earth). For more information, see our copyright guidelines: http://journals.plos.org/plosone/s/licenses-and-copyright.

1.    You may seek permission from the original copyright holder of Figure #1 to publish the content specifically under the CC BY 4.0 license. 

Reviewers' comments:

Reviewer's Responses to Questions

**Comments to the Author**

1. Is the manuscript technically sound, and do the data support the conclusions?

Reviewer #1: Yes

Reviewer #2: Yes

2. Has the statistical analysis been performed appropriately and rigorously? 

Reviewer #1: Yes

Reviewer #2: Yes

3. Have the authors made all data underlying the findings in their manuscript fully available?

Reviewer #1: Yes

Reviewer #2: Yes

4. Is the manuscript presented in an intelligible fashion and written in standard English?

Reviewer #1: No

Reviewer #2: Yes

5. Review Comments to the Author

Reviewer #1: In this study, the authors conduct population genetic analyses in an endangered plant species (Erysimum teretifolium) that displays a patchy distribution. It seems to be a resubmission of a previous manuscript submitted to the same journal. In addition to this new version, I have also read the previous round of review. While I agree with the former reviewers that the results of this study may not be groundbreaking, I do agree with the authors that this type of studies (with important implications for conservation) must be published. If I am not wrong, this journal does not reject potential contributions on the basis of originality alone.

According to the cover letter, the authors have included new genomic data in this resubmission, which provides an interesting pattern between microsatellite (I also must agree it is unfortunately quite limited) and plastome / ITS data. My main concern, indeed, is not the quantity of data, but the way the background of the study is presented. The authors made an interesting parallel between island distributions and species inhabiting island-like habitats. This is, in my view, the best way to present this study, but I do believe that the paper would be much more attractive to plant biologists and “island people”, in general, if the references and concepts are updated and the conceptual framework revised. I also provide some comments to improve readability, and a few ideas that may be helpful for data analysis (at the authors´ discretion). I hope the following points may help them in providing a more focused manuscript but, providing some areas are improved, I do support its publication in PLoSONE:

L62-65: One key factor in explaining such contrasting results is the consideration of fruit type (or fruit dispersal syndromes): for plant species for which animals act as dispersal vectors, differences among islands (patches, in the present study) would tend to be smaller than in plant species with other means of dispersal (dry fruits, in general; García-Verdugo et al. Bot J Linn Soc 2014). I think this point is worth mentioning early in the Introduction since Erysimum would fit in the second category, and would therefore make predictions more clear (L158-160).

L71-73: The review by Itescu (Ecography 2019) would probably reduce the number of citations needed to convey this relevant idea.

L81-83: I do not seem to completely understand the intention of this statement. On the other hand, it is supported by quite old citations; there are more recent reviews on the patterns of genetic diversity as a function of life history traits, including Nybom Mol Ecol 2004, or in the case of islands, Stuessy et al. Bot J Linn Soc 2014, García-Verdugo et al. Bot J Linn Soc 2014.

L84-85: I refrain from spreading these classic ideas for which recent studies are providing strong evidence that they only apply to very specific lineages/island conditions (see for instance García-Verdugo et al. J Biogeogr 2017; Burns J Biogeogr 2018; García-Verdugo et al. Ann Bot 2019)

L90. This is particularly true when dealing with woody (long-lived) taxa. However, some more recent citations on the relationship between population genetic diversity and species/population traits would be in order (e.g. Holsinger & Weir Nature Rev Gen 2009; Ellegren & Galtier Nature Rev Gen 2016).

L187. When describing E. capitatum, the ploidy level of this species should be specified as well.

L244. Microsatellite analysis. I am missing much more justification on why such a small number of loci were assayed (problems with transferability with other loci, allele variation difficult to interpret?)

L268. How many were these? Because your microsatellite dataset is not large, it is very important to describe well anything related to the quality of such data.

L270. Since you produced pedigree data using SSRs, allele dosage might be inferred from these (see Dufresne et al. Mol Ecol 2014). This would provide new opportunities to extract information from the SSR dataset.

L310. A histogram showing the result of the flow cytometry analysis could be nice as supplementary information.

L386-390 and L514 ff. In my view (and experience), successful PCR amplification of nSSR loci does not guarantee that all of the alleles of the sample/population have been amplified (just to cite a popular review, see Dakin & Avise Heredity 2004); sequence mutations linked with particular alleles reduces the number of allele variants in a heterozygote, which is particularly difficult to detect in polyploids. I do understand that applying typical software to test for the presence of null alleles is not a feasible option for these polyploids, but I would avoid using imprecise statements for justification. I would try to explain instead why the presence of null alleles would not compromise the main conclusions of the study.

Discussion: I would start this fundamental part of the manuscript with one strong statement, rather than emphasizing methodological aspects of the study. In my view, some of the first paragraphs could be transferred to Results, but this may be a matter of taste.

Minor points

Abstract (and throughout the text, e.g. L151) “microsatellites”: please be specific with regard to the type of genome (I assume they are nuclear, but this should be clearly indicated)

Abstract. “however when the BD population was removed”: I think it should read “however, when one outlier population was removed” or briefly explain why this population blurs the general pattern; i.e. please do not refer to one population code that the reader is not familiar with.

Abstract. “pervasive admixture”: “extensive”, “substantial” instead of “pervasive” sounds a bit better to me

L68: The beginning of this sentence sounds somewhat colloquial. What about “This model predicts…” instead?

L100-103. Please provide scientific names for these examples; L199 same for Ponderosa pine

L291. Replace “mt genome” with “mitogenome” or “mitochondrial genome”

Table 1. When referring to “microsatellite fragments”, I assume it means “microsatellite loci” (or “microsatellite alleles”?)

Table 2. AMOVA results are usually displayed the other way round (i.e. from upper to lower hierarchical levels), but the information is perfectly understandable in the present form (the number of decimal digits could be reduced, though).

C. García-Verdugo

Reviewer #2: This article uses many different molecular markers and analytic methods to explore the population genetic structure of sandhill endemic Erysimum teretifolium, and explores the impact of the mating system and island-like distribution on the population genetic structure of this species. In general, this article is quite well written, but throughout the article, I feel that it seems to deliberately emphasise the influence of the mating system on its genetic structure. However, the authors do not directly observe the reproductive mode or have any adequate experimental design on the mating system. Notably, in self-incompatible species, it is assumed that all individuals can only accept gametes of different genotypes, and the differentiation between populations will not be apparent. Therefore, although their geographical distribution is patches, they will not necessarily present an island model. The authors can use the island model as a null model, instead of “to muster supporting evidences for a foregone conclusion”. Apart from that, I have no other further comments. As I mentioned before, I think this is a well-written paper. I think the problems described above can be modified in the writing manner.

6. PLOS authors have the option to publish the peer review history of their article (what does this mean?). If published, this will include your full peer review and any attached files.

Reviewer #1: No

Reviewer #2: No

---

## [Author Response · Author response to Decision Letter 0]

25 Mar 2020

PONE-D-19-33086

Genome skimming and microsatellite analysis reveal contrasting patterns of genetic diversity in a rare sandhill endemic (Erysimum teretifolium, Brassicaceae)

PLOS ONE

Dear Dr. Whittall,

Thank you for submitting your manuscript to PLOS ONE. After careful consideration, we feel that it has merit but does not fully meet PLOS ONE’s publication criteria as it currently stands. Therefore, we invite you to submit a revised version of the manuscript that addresses the points raised during the review process.

We would appreciate receiving your revised manuscript by Apr 16 2020 11:59PM. To enhance the reproducibility of your results, we recommend that if applicable you deposit your laboratory protocols in protocols.io, where a protocol can be assigned its own identifier (DOI) such that it can be cited independently in the future. For instructions see: http://journals.plos.org/plosone/s/submission-guidelines#loc-laboratory-protocols

• A rebuttal letter that responds to each point raised by the academic editor and reviewer(s). This letter should be uploaded as separate file and labeled 'Response to Reviewers'.

• A marked-up copy of your manuscript that highlights changes made to the original version. This file should be uploaded as separate file and labeled 'Revised Manuscript with Track Changes'.

• An unmarked version of your revised paper without tracked changes. This file should be uploaded as separate file and labeled 'Manuscript'.

We look forward to receiving your revised manuscript.

Kind regards,

Kuo-Hsiang Hung

Academic Editor

PLOS ONE

Journal Requirements:

Done.

2. We noted in your submission details that a portion of your manuscript may have been presented or published elsewhere. ["This manuscript is an improved version of a thesis presented by JAH in partial fulfillment of requirements for the SCU Honors Program. “Team Wallflower,” especially Miranda Melen, was instrumental to the completion of this project. Dena Grossenbacher and the reviewers provided helpful comments."] Please clarify whether this publication was peer-reviewed and formally published. If this work was previously peer-reviewed and published, in the cover letter please provide the reason that this work does not constitute dual publication and should be included in the current manuscript.

No portion of the manuscript has been published elsewhere. We modified the acknowledgements section to avoid any confusion. It now reads, “The authors are very grateful to “Team Wallflower” and especially Miranda Melen for her invaluable help throughout this study. JAH was supported by an ALZA Corporation Scholarship. Dena Grossenbacher and the anonymous reviewers provided helpful comments on an earlier draft.”

Thanks for noticing. We have removed this phrase in the manuscript since this information is not relevant for the paper. 

4. We note that Figure #1 in your submission contains satellite images which may be copyrighted. All PLOS content is published under the Creative Commons Attribution License (CC BY 4.0), which means that the manuscript, images, and Supporting Information files will be freely available online, and any third party is permitted to access, download, copy, distribute, and use these materials in any way, even commercially, with proper attribution. For these reasons, we cannot publish previously copyrighted maps or satellite images created using proprietary data, such as Google software (Google Maps, Street View, and Earth). For more information, see our copyright guidelines: http://journals.plos.org/plosone/s/licenses-and-copyright.

1. You may seek permission from the original copyright holder of Figure #1 to publish the content specifically under the CC BY 4.0 license. 

We have replaced the GoogleMaps protected image with a map that is freely available from USGS National Map Viewer (thanks for the recommendation). We have included appropriate source information in the figure caption and a scale bar.

Reviewers' comments:

Reviewer's Responses to Questions

Comments to the Author

1. Is the manuscript technically sound, and do the data support the conclusions?

Reviewer #1: Yes

Reviewer #2: Yes

2. Has the statistical analysis been performed appropriately and rigorously? 

Reviewer #1: Yes

Reviewer #2: Yes

3. Have the authors made all data underlying the findings in their manuscript fully available?

Reviewer #1: Yes

Reviewer #2: Yes

4. Is the manuscript presented in an intelligible fashion and written in standard English?

Reviewer #1: No

Reviewer #2: Yes

We have followed the reviewer suggestions and carefully revised the manuscript to improve the language. We hope now the manuscript is free of grammatical or typographical errors. 

5. Review Comments to the Author

Reviewer #1: In this study, the authors conduct population genetic analyses in an endangered plant species (Erysimum teretifolium) that displays a patchy distribution. It seems to be a resubmission of a previous manuscript submitted to the same journal. In addition to this new version, I have also read the previous round of review. While I agree with the former reviewers that the results of this study may not be groundbreaking, I do agree with the authors that this type of studies (with important implications for conservation) must be published. If I am not wrong, this journal does not reject potential contributions on the basis of originality alone.

According to the cover letter, the authors have included new genomic data in this resubmission, which provides an interesting pattern between microsatellite (I also must agree it is unfortunately quite limited) and plastome / ITS data. My main concern, indeed, is not the quantity of data, but the way the background of the study is presented. The authors made an interesting parallel between island distributions and species inhabiting island-like habitats. This is, in my view, the best way to present this study, but I do believe that the paper would be much more attractive to plant biologists and “island people”, in general, if the references and concepts are updated and the conceptual framework revised. I also provide some comments to improve readability, and a few ideas that may be helpful for data analysis (at the authors´ discretion). I hope the following points may help them in providing a more focused manuscript but, providing some areas are improved, I do support its publication in PLoSONE:

We are very grateful for all these valuable suggestions, which have definitely helped us improve the manuscript. We did our best to address all of them.

L62-65: One key factor in explaining such contrasting results is the consideration of fruit type (or fruit dispersal syndromes): for plant species for which animals act as dispersal vectors, differences among islands (patches, in the present study) would tend to be smaller than in plant species with other means of dispersal (dry fruits, in general; García-Verdugo et al. Bot J Linn Soc 2014). I think this point is worth mentioning early in the Introduction since Erysimum would fit in the second category, and would therefore make predictions more clear (L158-160).

Thanks for noticing. Following this suggestion, we stressed early in the introduction the importance of the fruit type for the genetic variability of islands in paragraphs 1 and 3.

L71-73: The review by Itescu (Ecography 2019) would probably reduce the number of citations needed to convey this relevant idea.

We were not aware of this new review paper. We found it very interesting and relevant. Thus, we have included it in the second paragraph of the Introduction. It now reads “However, investigations into the distribution of genetic variation in island-like habitats are relatively rare given the diversity of naturally patchy, edaphically unique habitats (reviewed in [11]).”

L81-83: I do not seem to completely understand the intention of this statement. On the other hand, it is supported by quite old citations; there are more recent reviews on the patterns of genetic diversity as a function of life history traits, including Nybom Mol Ecol 2004, or in the case of islands, Stuessy et al. Bot J Linn Soc 2014, García-Verdugo et al. Bot J Linn Soc 2014.

We agree this sentence could lead to misunderstandings. We modified this phrase to clear things up and we replaced the old citations with Reviewer #1’s suggestions. The topic sentence of paragraph 3 now reads, “The genetic predictions for species on islands and in island-like habitats must also account for life-history traits that can directly affect gene flow, such as mating system [9,13] and seed dispersal [14]. In terms of mating system,…”

L84-85: I refrain from spreading these classic ideas for which recent studies are providing strong evidence that they only apply to very specific lineages/island conditions (see for instance García-Verdugo et al. J Biogeogr 2017; Burns J Biogeogr 2018; García-Verdugo et al. Ann Bot 2019)

Thanks you for noticing. Following your recommendation, we have revised this sentence to avoid any controversy regarding the dispersal capabilities of island species since we want to focus on the importance of mating system in predicting the distribution of genetic diversity within and among populations. It now reads, “In terms of mating system, many island species are self-compatible [15], which greatly affect the distribution of genetic diversity [16]. ”

L90. This is particularly true when dealing with woody (long-lived) taxa. However, some more recent citations on the relationship between population genetic diversity and species/population traits would be in order (e.g. Holsinger & Weir Nature Rev Gen 2009; Ellegren & Galtier Nature Rev Gen 2016).

We updated the references following your suggestions.

L187. When describing E. capitatum, the ploidy level of this species should be specified as well.

Thanks for noticing. We added the ploidy level of E. capitatum at the end of the Sampling paragraph in the Methods. It now reads, “This species, as with the all the taxa in the E. capitatum alliance, is also a hexaploid (2n = 36; [29]).”

L244. Microsatellite analysis. I am missing much more justification on why such a small number of loci were assayed (problems with transferability with other loci, allele variation difficult to interpret?)

We agree that this needs justification. We have explained how we arrived at four microsatellite loci in the topic sentence of the “Microsatellite methodology” subsection with the following sentence, “After an initial survey of the 10 nuclear microsatellite loci developed from the European E. mediohispanicum [43], we selected four of the most promising loci for further analysis.”

L268. How many were these? Because your microsatellite dataset is not large, it is very important to describe well anything related to the quality of such data.

There were four out of 24 markers that appeared in F1 offspring, but were not present in the parents. We have included this information at the end of the second paragraph of the “Microsatellite methodology” subsection. The sentence now reads, “Four fragments that appeared in offspring that were not present in the parents were removed from all subsequent microsatellite analyses.”

L270. Since you produced pedigree data using SSRs, allele dosage might be inferred from these (see Dufresne et al. Mol Ecol 2014). This would provide new opportunities to extract information from the SSR dataset.

We looked for dosage in our segregating population and it wasn’t immediately obvious. Thank you for reminding us about the Dufresne et al. 2014 review of molecular markers in polyploids. My read on it is less hopeful than the reviewer’s suggests. In the microsatellite section, they write, “In polyploids, inability to reliably utilize codominant scoring reduces the usefulness of microsatellites relative to diploids and to AFLPs.” (p.44). They describe the rare successful application of microsatellites as codominant markers in polyploids following dosage determination from a segregation analysis (e.g., Esselink et al. 2004; Landergott et al. 2006; and Luttikhuizen et al. 2007), yet dosage determination in our application are complicated by the heterospecific nature of the microsats to begin with and then compounded with the well-known microsatellite stutter. We are much more confident in our analysis using the microsatellite bands treated as dominant markers following precedent (Saltonstall 2003, Andreakis et al. 2009, and even your own study Garcia-Verdugo et al. 2013).

L310. A histogram showing the result of the flow cytometry analysis could be nice as supplementary information.

Although the values were very consistent (2.82 – 3.06), there were only four individuals sampled. The four measurements per sample only represent technical replication and would be misleading to report as natural variation in a histogram with error bars. Therefore, it would be a histogram with four bars and no biological error. We prefer to report the mean and the range since the spread is so limited. We have chosen not to create the recommended histogram, even as Supplementary Information.

L386-390 and L514 ff. In my view (and experience), successful PCR amplification of nSSR loci does not guarantee that all of the alleles of the sample/population have been amplified (just to cite a popular review, see Dakin & Avise Heredity 2004); sequence mutations linked with particular alleles reduces the number of allele variants in a heterozygote, which is particularly difficult to detect in polyploids. I do understand that applying typical software to test for the presence of null alleles is not a feasible option for these polyploids, but I would avoid using imprecise statements for justification. I would try to explain instead why the presence of null alleles would not compromise the main conclusions of the study.

In our Results, we have qualified our method for testing for null alleles in the subsection on “Microsatellite variation” using the more cautious descriptor, “- this is just one method of identifying null alleles for nuclear microsatellites (see [54] for a comprehensive review).”

In the Discussion of the “Microsatellite analysis”, we have toned down the interpretation with the phrase “null alleles appear to be rare…” followed by a paraphrased version of the Reviewer’s concise statement, “However, our power to detect null alleles was limited to PCR reactions that produced no fragments – we would not have detected missing fragments as long as one allele amplified. Mutations at or near the priming sites linked with particular alleles can reduce the number of allele variants in a heterozygote, which can be particularly difficult to detect in polyploids [54].’

Discussion: I would start this fundamental part of the manuscript with one strong statement, rather than emphasizing methodological aspects of the study. In my view, some of the first paragraphs could be transferred to Results, but this may be a matter of taste.

Good idea. We have summarized our findings to start the Discussion section before going into the Genome skimming and microsatellite sections. It reads, “Genome skimming yielded a nearly complete chloroplast genome that was deeply divided among the E. teretifolium populations sampled, yet very little variation was detected in the nuclear ribosomal cistron.. In contrast, the nuclear microsatellite analysis indicated the majority of genetic variation was found within populations with limited (yet significant) population differentiation.”

Minor points

Abstract (and throughout the text, e.g. L151) “microsatellites”: please be specific with regard to the type of genome (I assume they are nuclear, but this should be clearly indicated). 

Yes, they are nuclear microsatellites. We clarified this throughout the manuscript..

Abstract. “however when the BD population was removed”: I think it should read “however, when one outlier population was removed” or briefly explain why this population blurs the general pattern; i.e. please do not refer to one population code that the reader is not familiar with. 

We modified this sentence in the Abstract according to the referee’s suggestion. It now reads, “however when one outlier population was removed from the analysis due to uncertain provenance…”

Abstract. “pervasive admixture”: “extensive”, “substantial” instead of “pervasive” sounds a bit better to me.

We used the word “substantial” in this sentence instead of “pervasive”, as suggested.

L68: The beginning of this sentence sounds somewhat colloquial. What about “This model predicts…” instead?

Thanks for noticing. We changed the beginning of the sentence as suggested. It now reads “This model is based on geographically distinct populations separated by barriers to gene flow but can also be applied to continental habitats that are island-like.”

L100-103. Please provide scientific names for these examples; L199 same for Ponderosa pine

We included the scientific names of plants and animals used as examples of the biota adapted to this habitat.

L291. Replace “mt genome” with “mitogenome” or “mitochondrial genome”

Done.

Table 1. When referring to “microsatellite fragments”, I assume it means “microsatellite loci” (or “microsatellite alleles”?)

Throughout the manuscript, we refer to the microsatellite data as “fragments” since they were treated as dominant markers. Referring to them as loci may confuse the reader that we used 24 different primer pairs. Furthermore, referring to them as alleles sounds like typical co-dominant microsats in which one can detect heterozygotes. If “fragments” sounds unusual, that was our intention to ensure the reader understood that they were being treated as dominant markers. To ensure clarity, we have defined what we mean by “fragments” upon first appearance in the text (Methods, “Microsatellite methodology”) where we state, “To confirm the reliability of our microsatellite data (hereafter referred to as “fragments” because they were treated as dominant markers)…”

Table 2. AMOVA results are usually displayed the other way round (i.e. from upper to lower hierarchical levels), but the information is perfectly understandable in the present form (the number of decimal digits could be reduced, though).

Thank you for the suggestions. Now the AMOVA table displays from upper (among groups) to lower (within population) hierarchical levels.

C. García-Verdugo

Reviewer #2: This article uses many different molecular markers and analytic methods to explore the population genetic structure of sandhill endemic Erysimum teretifolium, and explores the impact of the mating system and island-like distribution on the population genetic structure of this species. In general, this article is quite well written, but throughout the article, I feel that it seems to deliberately emphasise the influence of the mating system on its genetic structure. However, the authors do not directly observe the reproductive mode or have any adequate experimental design on the mating system. Notably, in self-incompatible species, it is assumed that all individuals can only accept gametes of different genotypes, and the differentiation between populations will not be apparent. Therefore, although their geographical distribution is patches, they will not necessarily present an island model. The authors can use the island model as a null model, instead of “to muster supporting evidences for a foregone conclusion”. Apart from that, I have no other further comments. As I mentioned before, I think this is a well-written paper. I think the problems described above can be modified in the writing manner.

We have made substantial changes to the “writing manner” throughout the manuscript based on this reviewer’s comment and the more extensive comments of Reviewer #1 and the Editor. We thank Reviewer #2 for noticing our deliberate references to the mating system of E. teretifolium. In a previous study, we experimentally determined that it is self-incompatible and relies on a diversity of pollinators for impressive levels of seed-set (see Melen et al. American Journal of Botany 2016). To clarify our approach, in the present study under review, we did not intend to expand our study of the mating system, only to use the previous results combined with the island-like habitat to examine the distribution of genetic variation. 

The foregone conclusion that “differentiation between populations will not be apparent” may not be assumed by all readers since these xeric habitats are isolated from one another by up to 7 km of unoccupiable habitat of mesic forest. The reviewer’s perspective that mating system determines the distribution of genetic diversity regardless of island-like habitats may not be shared by all – Reviewer #1 even applauds our set-up with these two options as alternatives with distinct patterns of genetic variation. 

The degree of population differentiation (even in self-incompatible species) relies on the geographic distance among populations, levels of gene flow (through both pollen as mentioned by Reviewer #2, but also by seed dispersal as mentioned by Reviewer #1), geological and climate history that would determine the degree of isolation among populations, and numerous other parameters that we cannot fathom. If the populations have been isolated for a long time and there has been very little gene flow, even a self-incompatible species can become differentiated (at least for nearly neutral markers like microsatellites and the chloroplast genome).

6. PLOS authors have the option to publish the peer review history of their article (what does this mean?). If published, this will include your full peer review and any attached files.

Do you want your identity to be public for this peer review? For information about this choice, including consent withdrawal, please see our Privacy Policy.

Reviewer #1: No

Reviewer #2: No

---

## [Decision Letter · Decision Letter 1]

17 Apr 2020

PONE-D-19-33086R1

Genome skimming and microsatellite analysis reveal contrasting patterns of genetic diversity in a rare sandhill endemic (Erysimum teretifolium, Brassicaceae)

PLOS ONE

Dear Dr. Whittall,

Thank you for submitting your manuscript to PLOS ONE. After careful consideration, we feel that it has merit but does not fully meet PLOS ONE’s publication criteria as it currently stands. Therefore, we invite you to submit a revised version of the manuscript that addresses the points raised during the review process.

We would appreciate receiving your revised manuscript by Jun 01 2020 11:59PM. To enhance the reproducibility of your results, we recommend that if applicable you deposit your laboratory protocols in protocols.io, where a protocol can be assigned its own identifier (DOI) such that it can be cited independently in the future. For instructions see: http://journals.plos.org/plosone/s/submission-guidelines#loc-laboratory-protocols

We look forward to receiving your revised manuscript.

Kind regards,

Kuo-Hsiang Hung

Academic Editor

PLOS ONE

Reviewers' comments:

Reviewer's Responses to Questions

**Comments to the Author**

1. If the authors have adequately addressed your comments raised in a previous round of review and you feel that this manuscript is now acceptable for publication, you may indicate that here to bypass the “Comments to the Author” section, enter your conflict of interest statement in the “Confidential to Editor” section, and submit your "Accept" recommendation.

Reviewer #1: All comments have been addressed

Reviewer #3: (No Response)

2. Is the manuscript technically sound, and do the data support the conclusions?

Reviewer #1: Yes

Reviewer #3: Partly

3. Has the statistical analysis been performed appropriately and rigorously? 

Reviewer #1: Yes

Reviewer #3: Yes

4. Have the authors made all data underlying the findings in their manuscript fully available?

Reviewer #1: Yes

Reviewer #3: Yes

5. Is the manuscript presented in an intelligible fashion and written in standard English?

Reviewer #1: Yes

Reviewer #3: Yes

6. Review Comments to the Author

Reviewer #1: I have revised the detailed cover letter and the new version of the manuscript, and the authors have addressed all the points raised in my review, providing a reasonable reply when my suggestions have not been followed. I particularly acknowledge the document with the track-changes activated, which greatly facilitates the revision. I fully support publication of the manuscript in its present form.

Reviewer #3: Del Valle et al. studied 8 island-like populations of Erysimum teretifolium and found significant effect of isolation by distance. The phylogenies were reconstructed by cpDNA and nrDNA and revealed two groups with high bootstrap support. Comprehensive microsatellite analyses were conducted, two of the populations were suggested for conservation. The manuscript is a revised version. It is well-written and most of the patterns are discussed in detail. However, there were still a few points need to be clarified. I listed these points as follows.

1. The phylogenies of cpDNA and nrDNA indicated different grouping of these populations, possibly caused by inconsistent maternal and paternal histories. Also, both trees did not support affinity between QH and BD. Need more clarification for these.

2. Structure result at K=6 indicated HWY and SHGW possessing most distinct genetic components, and the authors suggested these two populations are important targets for conservation. Generally I agree with this point. By the way, the result at K=2 indicated SHGW and QH possessing distinct components while the other populations being admixtures of them. According to the location of populations and the dispersal direction suggested by wind (as mentioned from northwestern), it seems to me that the QH is an important population in this sense. Moreover, the information that BD might be a clonal population by QH also support this. At the result of K=6, although the identity of QH is blurred by multiple genetic components, I still suggested this population could be further discussed.

3. The map of population distribution (Fig 1B) needs an indicator for north.

7. PLOS authors have the option to publish the peer review history of their article (what does this mean?). If published, this will include your full peer review and any attached files.

Reviewer #1: Yes: Carlos García-Verdugo

Reviewer #3: No

---

## [Author Response · Author response to Decision Letter 1]

18 Apr 2020

PONE-D-19-33086R1

Rebuttal Letter Accompanying 2nd Revision of:

“Genome skimming and microsatellite analysis reveal contrasting patterns of genetic diversity in a rare sandhill endemic (Erysimum teretifolium, Brassicaceae)”

Review Comments to the Author

Reviewer #1: I have revised the detailed cover letter and the new version of the manuscript, and the authors have addressed all the points raised in my review, providing a reasonable reply when my suggestions have not been followed. I particularly acknowledge the document with the track-changes activated, which greatly facilitates the revision. I fully support publication of the manuscript in its present form.

Reviewer #3: Del Valle et al. studied 8 island-like populations of Erysimum teretifolium and found significant effect of isolation by distance. The phylogenies were reconstructed by cpDNA and nrDNA and revealed two groups with high bootstrap support. Comprehensive microsatellite analyses were conducted, two of the populations were suggested for conservation. The manuscript is a revised version. It is well-written and most of the patterns are discussed in detail. However, there were still a few points need to be clarified. I listed these points as follows.

1. The phylogenies of cpDNA and nrDNA indicated different grouping of these populations, possibly caused by inconsistent maternal and paternal histories. Also, both trees did not support affinity between QH and BD. Need more clarification for these.

Although Figure 2A (cpDNA) and 2B (nrDNA) show different topologies as noted by the reviewer, the nrDNA phylogenetic analysis shows no branches within the ERTE clade with bootstrap support >50%. Thus, there is no incongruence between the cpDNA and nrDNA phylogenies since the nrDNA tree only supports the monophyly of ERTE to the exclusion of the ERCAAN samples. A discussion about maternal and paternal evolutionary histories would suggest the nrDNA topology is strongly supported. We tried to improve this message in the Genome Skimming portion of the Results entitled, “Analysis of the nrDNA alignment” where we state: “Maximum likelihood phylogenetic analysis using RAxML produced a tree with only a single well-supported branch defining the monophyly of the ERTE samples as separate from the two ERCAAN samples (BS = 100%; Fig 2B).” To help clarify this previously stated result, we have added a clarifying sentence to the last paragraph in the Discussion section on “Genome skimming…” that now states, “Although the cpDNA tree topology appears incongruent with that of the nrDNA region, the latter lacks any branches within ERTE with bootstrap values > 50%.”

Thank you for pointing out the inconsistency between the cpDNA tree (Fig. 2A) and the microsatellite affinities (e.g., Fig. 4B), especially with regard to QH and BD where seeds may have been moved unofficially. We now raise this point in the Discussion section entitled “Island-like patterns…” where we previously discussed the possibility of “unofficial trafficking” of seeds between QH and BD in regards to the microsatellite data alone. The statement is now followed by this phrase, “however BD and QH fall into separate, strongly supported cpDNA lineages (Fig. 2A).” We have added a comparable qualifying statement in the second paragraph of the Discussion section entitled, “Conservation implications”.

2. Structure result at K=6 indicated HWY and SHGW possessing most distinct genetic components, and the authors suggested these two populations are important targets for conservation. Generally I agree with this point. By the way, the result at K=2 indicated SHGW and QH possessing distinct components while the other populations being admixtures of them. According to the location of populations and the dispersal direction suggested by wind (as mentioned from northwestern), it seems to me that the QH is an important population in this sense. Moreover, the information that BD might be a clonal population by QH also support this. At the result of K=6, although the identity of QH is blurred by multiple genetic components, I still suggested this population could be further discussed.

We agree that in the K=2 Structure analysis of the microsatellite data, the most genetically divergent populations are SHGW and QH (and BD looks very similar to QH). Thank you for pointing out the need to include QH as a priority for preserving the remaining genetic variation based on the microsatellite results.

We have emphasized the evidence for the genetic similarity of QH and BD revealed by the K=2 Structure analysis in the Discussion section of “Island-like patterns…” in the 1st full paragraph. It now reads (underlined = new), “Structure analysis of the microsatellite data (K = 2) suggests BD and QH are very similar genetically – most individuals from both populations have the same proportions of admixture (Fig 4B).” Furthermore, we have added QH to the list of populations with unique genetic variation. The sentence in the “Conservation implications” section of the Discussion now read, “Yet, in order to capture the remaining genetic variation, peripheral populations (such as QH, HWY, and SHGW) must also be preserved.”

3. The map of population distribution (Fig 1B) needs an indicator for north.

 Done.

---

## [Decision Letter · Decision Letter 2]

29 Apr 2020

Genome skimming and microsatellite analysis reveal contrasting patterns of genetic diversity in a rare sandhill endemic (Erysimum teretifolium, Brassicaceae)

PONE-D-19-33086R2

Dear Dr. Whittall,

We are pleased to inform you that your manuscript has been judged scientifically suitable for publication and will be formally accepted for publication once it complies with all outstanding technical requirements.

With kind regards,

Kuo-Hsiang Hung

Academic Editor

PLOS ONE

Additional Editor Comments (optional):

Reviewers' comments:

Reviewer's Responses to Questions

**Comments to the Author**

1. If the authors have adequately addressed your comments raised in a previous round of review and you feel that this manuscript is now acceptable for publication, you may indicate that here to bypass the “Comments to the Author” section, enter your conflict of interest statement in the “Confidential to Editor” section, and submit your "Accept" recommendation.

Reviewer #1: All comments have been addressed

Reviewer #3: All comments have been addressed

2. Is the manuscript technically sound, and do the data support the conclusions?

Reviewer #1: Yes

Reviewer #3: Yes

3. Has the statistical analysis been performed appropriately and rigorously? 

Reviewer #1: Yes

Reviewer #3: Yes

4. Have the authors made all data underlying the findings in their manuscript fully available?

Reviewer #1: Yes

Reviewer #3: Yes

5. Is the manuscript presented in an intelligible fashion and written in standard English?

Reviewer #1: Yes

Reviewer #3: Yes

6. Review Comments to the Author

Reviewer #1: (No Response)

Reviewer #3: (No Response)

7. PLOS authors have the option to publish the peer review history of their article (what does this mean?). If published, this will include your full peer review and any attached files.

Reviewer #1: No

Reviewer #3: No

---

## [Editor Report · Acceptance letter]

5 May 2020

PONE-D-19-33086R2 

Genome skimming and microsatellite analysis reveal contrasting patterns of genetic diversity in a rare sandhill endemic (*Erysimum teretifolium*, Brassicaceae) 

Dear Dr. Whittall:

I am pleased to inform you that your manuscript has been deemed suitable for publication in PLOS ONE. Congratulations! Your manuscript is now with our production department. 

With kind regards,

on behalf of

Dr. Kuo-Hsiang Hung 

Academic Editor

PLOS ONE